# Cryo-EM unveils kinesin KIF1A's processivity mechanism and the impact of its pathogenic variant P305L

Matthieu P. M. H. Benoit [1,2] ✉, Lu Rao [1,2], Ana B. Asenjo [1], Arne Gennerich [1] ✉ & Hernando Sosa [1] ✉

Mutations in the microtubule-associated motor protein KIF1A lead to severe neurological conditions known as KIF1A-associated neurological disorders (KAND). Despite insights into its molecular mechanism, high-resolution structures of KIF1A-microtubule complexes remain undefined. Here, we present 2.7-3.5 Å resolution structures of dimeric microtubule-bound KIF1A, including the pathogenic P305L mutant, across various nucleotide states. Our structures reveal that KIF1A binds microtubules in one- and two-heads-bound configurations, with both heads exhibiting distinct conformations with tight inter-head connection. Notably, KIF1A's class-specific loop 12 (K-loop) forms electrostatic interactions with the C-terminal tails of both α- and β-tubulin. The P305L mutation does not disrupt these interactions but alters loop-12's conformation, impairing strong microtubule-binding. Structure-function analysis reveals the K-loop and head-head coordination as major determinants of KIF1A's superprocessive motility. Our findings advance the understanding of KIF1A's molecular mechanism and provide a basis for developing structure-guided therapeutics against KAND.

KIF1A, a neuron-specific member of the kinesin-3 family, is a microtubule (MT) plus-end-directed motor protein that plays a key role in the migration of nuclei in differentiating brain stem cells[1,2] and the transport of synaptic precursors and dense core vesicles to axon terminals[3–8]. The dysfunction of KIF1A is linked to a spectrum of severe neurodevelopmental and neurodegenerative diseases known as KIF1A-associated neurological disorders (KAND). These disorders include progressive spastic paraplegias, microcephaly, encephalopathies, intellectual disability, autism, autonomic and peripheral neuropathy, optic nerve atrophy, cerebral and cerebellar atrophy, and others[9–51]. To date, more than 145 inherited and de novo KAND mutations have been identified, and these mutations span the entirety of the KIF1A protein sequence[9]. The majority are located within the motor domain (MD or 'head')[9] and are thus predicted to affect the motor's motility properties whereas mutations located outside the motor domain are likely involved in mediating dimerization, autoinhibition, and/or cargo

binding[10]. Our own work[9,52] and the work of others[53,54] has indeed shown that KAND motor domain mutations affect the motor's ability to generate force and movement.

Through the patient advocacy group *KIF1A.org*, more than 580 families with children and adults with KIF1A mutations have come together to improve the lives of those affected by KAND and to accelerate drug discovery. Despite these efforts, the molecular etiologies of KAND remain poorly understood, in part because KIF1A's molecular mechanism remains unclear. For example, KIF1A is fast and super-processive[55–59] (the motor can take more than a thousand steps before dissociating), and at the same time, gives up easily under load[52]. These behaviors distinguish KIF1A from the founding member of the kinesin family, kinesin-1[60,61], and it remains unknown how KIF1A achieves this unique set of properties.

While the motor domain of KIF1A is highly conserved among the members of the kinesin superfamily[62–64], it also exhibits unique

[1]Department of Biochemistry and Gruss-Lipper Biophotonics Center, Albert Einstein College of Medicine, Bronx, NY 10461, USA. [2]These authors contributed equally: Matthieu P. M. H. Benoit, Lu Rao. ✉e-mail: matthieu.benoit@univ-rennes.fr; arne.gennerich@einsteinmed.edu; hernando.sosa@einsteinmed.edu

structural elements that contribute to its distinct functional properties. Notably, KIF1A contains a highly positively charged loop (loop-12) known as the "K-loop" due to its KNKKKKK sequence[65]. Weak electrostatic interaction between the K-loop and the C-terminal tubulin tails facilitate the biased diffusion of monomeric KIF1A towards the MT-plus-end[65–68]. Initially, it was thought that this mechanism solely governed KIF1A motility. However, subsequent research revealed that cargo binding induces dimerization, enabling KIF1A dimers to move processively along MTs akin to conventional kinesin (kinesin-1)[59,69,70].

While the biased diffusion of a single motor domain may not represent the main physiological mode of KIF1A motility, the K-loop remains integral to the function of dimeric KIF1A. Several studies have demonstrated the significant contribution of the K-loop to KIF1A's characteristically long run lengths[57,71], a phenomenon referred to as superprocessivity[59]. Specifically, the K-loop enhances KIF1A's affinity for MTs during its weak-MT binding ADP state[66,72]. Furthermore, it facilitates the initial binding of KIF1A to MTs[72] and plays a role in the motor's rapid reattachment to the MT following detachment under load[52]. This reattachment ability results in a unique sawtooth-like force generation pattern[52].

Despite these insights, our understanding of KIF1A's unique motile properties remains limited due to the absence of high-resolution structural data of the KIF1A-MT complex. The KIF1A motor domain has been visualized under various experimental conditions using X-ray crystallography[73–77] and in complex with MTs at medium-resolution (6-15 Å) using cryoEM[67,73,78,79]. Importantly, there is no cryo-EM information on MT-bound KIF1A dimers, the functional oligomeric form required for unidirectional processive motility[59], especially with no structural data on MT-bound KIF1A in its two-heads-bound configuration. To fully understand the structural basis of KIF1A motility, high-resolution structures of KIF1A-MT complexes are needed. Such high-resolution data are essential to accurately trace the polypeptide chains, determine the position of side chains and differentiate between possible coexisting conformations. Furthermore, previous structural studies have largely left the functionally important K-loop unresolved, underscoring the necessity of high-resolution data not only for a comprehensive understanding of KIF1A's motility and force-generation mechanisms but also for advancing structure-guided drug discovery efforts, which hinge on detailed insights into this key structural element.

To address these challenges, we determined the high-resolution cryo-EM structures of dimeric KIF1A bound to MTs at 2.7–3.5 Å overall resolution in various conformational states and nucleotide conditions: with the non-hydrolysable ATP analog, AMP-PNP (adenylyl-imidodiphosphate), with ADP, and without nucleotides (Apo-state). These structures represent different stages of the KIF1A-MT ATPase cycle. Our results highlight the full structure of the K-loop and its interaction with the C-terminal tails of both α- and β-tubulin, providing structural evidence for the K-loop's pivotal role in KIF1A superprocessivity. Moreover, we show structurally and functionally that the coordination between KIF1A's two motor domains contributes to the processive motion of KIF1A. Additionally, we have determined the near-atomic resolution structure of the P305L KAND mutant bound to MTs, providing structural insights into how this mutation impairs KIF1A functionality.

In summary, our work provides critical insights into KIF1A's mechanism and demonstrates the possibility of obtaining high-resolution structures of MT-bound KAND mutants, which could inform the development of targeted therapies based on these mutant structures.

## Results

### High-resolution structures of the KIF1A-MT complexes

To elucidate the structural basis of KIF1A's unique motile properties, we obtained high-resolution cryo-EM structures of dimeric KIF1A bound to microtubules (MTs) in various nucleotide states, representing different stages of KIF1A's ATPase cycle (Fig. 1, Supplementary Fig. 2). We adapted a local classification and refinement strategy to the cryo-EM data[80], which revealed distinct conformations of the two motor domains when associated with MTs (Supplementary Figs. 3, 4). The cryo-EM maps achieved an overall resolution of 2.7–3.5 Å (Supplementary Figs. 5, 6).

In the presence of AMP-PNP (abbreviated as ANP in our maps and model names), we observed classes where KIF1A's two motor domains are connected by its coiled-coil dimerization domain and engaged with the core of two adjacent tubulin heterodimers along an MT protofilament (Fig. 1a). The leading head, positioned closer to the MT plus-end, features a backward-oriented neck-linker and an open nucleotide-binding pocket, whereas the trailing head exhibits a docked, forward-oriented neck-linker and a closed nucleotide-binding pocket. Both heads contain AMP-PNP bound in their nucleotide-binding sites (Figs. 1a and 2a and Supplementary Fig. 7). Other classes show single MT-bound heads with a trailing-like head and a missing leading head or with the trailing head of another dimer occupying the leading position (Supplementary Figs. 3 and 4). Our classification further revealed that the trailing head, whether in a two-heads-bound configuration or as a single-bound head with a docked neck-linker, adopts three different conformations (Supplementary. Fig. 4). Two classes, MT-KIF1A-ANP-$T_2L_1$ and MT-KIF1A-ANP-$T_3L_1$, shared strong similarities and were averaged to create the map depicted in Fig. 1a (MT-KIF1A-ANP-$T_{23}L_1$), from which we constructed the atomic model representing the average two-heads-bound configuration of KIF1A-ANP (Fig. 2a).

The most distinct conformation, termed T1, was exclusively observed in one-head-bound classes and is characterized by having a docked neck-linker but a nucleotide-binding pocket that is semi-closed (Fig. 2d, Supplementary Figs. 4, 8). This intermediate closure suggests that binding of the partner motor domain in the leading position may stabilize the fully closed conformation of the trailing head. This supports the notion of enhanced inter-head coordination, where the trailing head assumes a catalytically competent, closed conformation once its partner is bound to the MT in the leading and open conformation.

In conditions with ADP or without nucleotides (Apo state), only classes with single MT-bound heads were detected, lacking a visible connection to their partner head domains (Fig. 1b, c). This configuration represents a state where one head is MT-bound while the tethered partner head is unbound and mobile. Extra densities near the MT-bound motor in both Apo and ADP states hint that the unbound head partially explores the space behind the bound motor domain (Supplementary Fig. 9), as also suggested by the backward orientation of the linker's initial segment in these structures (Supplementary Fig. 9e, f). The structure of the MT-bound motor domain was similar in both conditions except for the presence or absence of ADP in the nucleotide-binding pocket (Figs. 1b, c and 2b, c). Notably, in the KIF1A ADP dataset, about 36% of the signal near the tubulin was unassigned, compared to a maximum of 5% in the other datasets (Supplementary Table 2). This significant portion of weak signal could represent weakly bound mobile motors, consistent with the expected lower MT affinity in the ADP state.

The open and closed conformations of the nucleotide-binding pocket in MT-bound kinesins[80,81] can be characterized by the distance between key residues involved in nucleotide binding (Fig. 2d and Supplementary Fig. 8). The largest and shortest distances values define, respectively, the open and closed conformations and intermediate distance values define the semi-closed (or semi-open) conformation. This conformation is observed in the crystal structures of MT-unbound free kinesins with ADP in the nucleotide-binding pocket[82], including for KIF1A (Fig. 2d)[73]. The open conformation we observed in high-resolution KIF1A-MT complexes, both in ADP and Apo states and in the leading head with bound AMP-PNP, represents, to

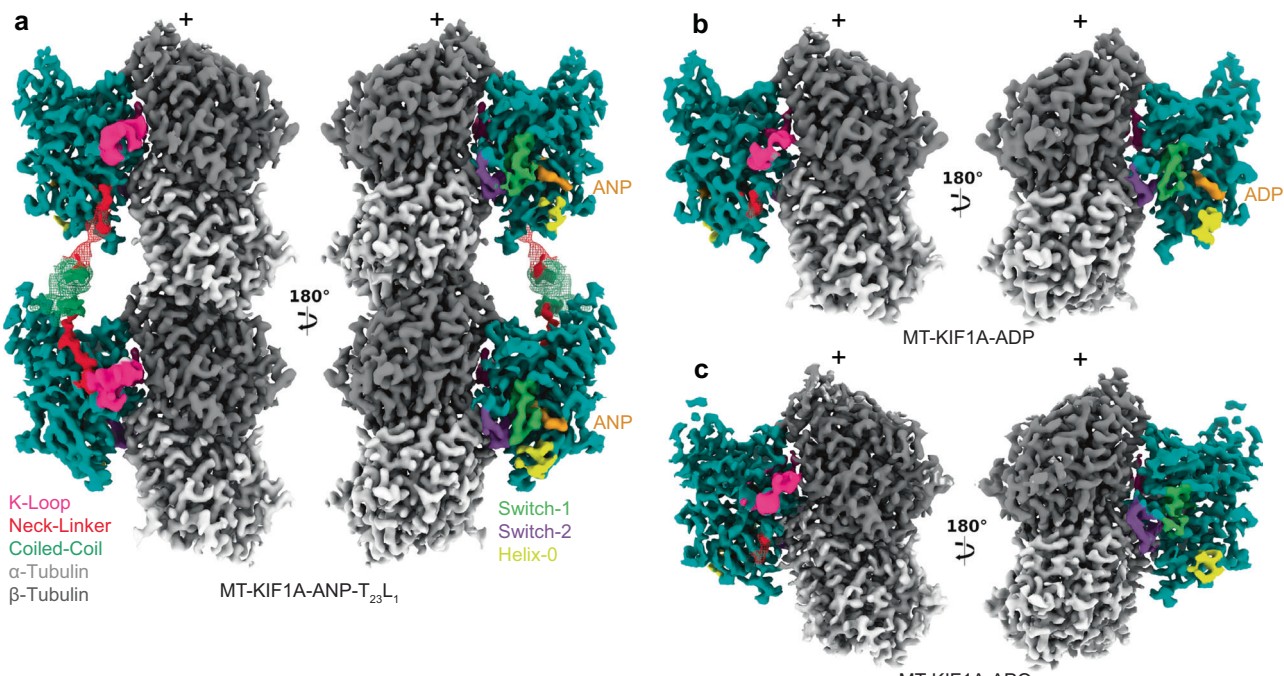

**Fig. 1 | Cryo-EM maps of microtubule-bound KIF1A in different nucleotide states.** Each panel shows two views of the isosurfaces of microtubule-bound KIF1A 3D maps, rotated 180° relative to each other. The surface colors emphasize different structural elements, as indicated in the figure labels. Map densities around the K-loop have been low-passed filtered and displayed at a lower contour level than the rest of the map for enhanced visualization of this mobile part. **a** Microtubule-bound KIF1A with the ATP analog AMP-PNP (abbreviated as ANP). The map corresponds to the average two-heads-bound configuration (MT-KIF1A-ANP-$T_{23}L_1$). The coiled-coil density and parts of the neck-linker from the leading head have been low-pass filtered and are displayed as a mesh at a lower contour level than that of the main map. Densities from the neck-linkers and coiled-coils connecting the two heads are visible. ANP densities are present in both the leading and trailing heads. Note: One-head-bound configurations were also found in the ANP datasets (Supplementary Figs. 3, 4, Supplementary Table 3). **b**, **c** Isosurface representations of KIF1A in the ADP and APO states (datasets MT-KIF1A-ADP and MT-KIF1A-APO). Isosurfaces are colored according to the corresponding structural element in the fitted atomic model, α- and β- tubulin in light and dark gray, respectively, the KIF1A motor domain in cyan with distinct structural elements (the K-loop, the neck-linker, the coiled-coil, the Switch-1, the Switch-2, and the bound nucleotide) colored as indicated in the insert.

the best of our knowledge, a novel conformation for the KIF1A motor domain. Previous X-ray crystallography-based models or models of KIF1A-MT complexes derived from lower resolution (>6 Å) cryo-EM data show KIF1A in either the closed or semi-closed conformation (Fig. 2d)[73–76,78,79]. The discernible differences between our findings and earlier cryo-EM results likely stem from the higher resolution of our cryo-EM data, which enables us to distinguish between open and semi-closed conformation that would otherwise appear similar at lower resolutions. Our comparison between the crystal structures and the high-resolution KIF1A-MT complexes shows that MT binding directly induces the nucleotide-binding pocket's transition from a semi-closed to an open conformation, providing a structural explanation for the enhanced ADP release associated with kinesin-MT binding[65,83–85].

**Conformation of the KIF1A neck-linker**

In the one-head-bound configuration of KIF1A observed in the presence of ADP and in the Apo state, the neck linker is undocked and extends backward towards the MT minus-end (Fig. 1 b, c, Supplementary Fig. 9e, f). Beyond the fourth neck-linker residue (N357), cryo-EM densities become notably weaker, indicating increased mobility in this region. In contrast, in the two-heads-bound configuration (MT-KIF1A-ANP-$T_{23}L_1$), clear cryo-EM densities reveal the neck-linkers of both trailing and leading heads, extending from the motor domain to the connecting coiled-coil dimerization domain (Figs. 1a, 3a). In this configuration, the neck-linker of the trailing head is docked onto the motor domain, while the neck-linker of the leading head remains undocked and oriented backward. This is also observed in the two-heads-bound MT complexes of KIF14[80], the only other high-resolution structure of a kinesin in such a configuration currently available.

However, distinct differences are evident in the KIF1A structure: the backward-oriented neck-linker of the leading head deviates more from the motor domain, taking a more direct, linear path between the two motor domains (Fig. 3a, b). These differences are likely due to the unique neck-linker sequences of KIF1A compared to other kinesins. Notably, KIF1A possesses a conserved proline at position 364, typical in the kinesin-3 family but absent in KIF14 and other kinesins (Fig. 3d). Within the two-heads-bound structure of KIF1A, this proline marks the start of an α-helix that pairs with a partner α-helix to form the coiled-coil dimerization domain (Fig. 3a). This feature results in a neck-linker for KIF1A that is shorter by approximately two to four residues compared to the neck-linkers of KIF14[80] or the kinesin-1 motor KIF5B[86] (Fig. 3b, c). Considering that the length of the neck-linker influences kinesin processivity[87], we hypothesize that the shorter neck-linkers of KIF1A in the two-heads-bound conformation facilitate tighter coordination between the catalytic cycles of the two motor domains, thereby enhancing processivity.

To test whether the presence of a proline residue at the end of the KIF1A neck-linker affects its structure and KIF1A processivity we created a KIF1A variant by replacing the conserved proline residue with leucine (P364L). This substitution mirrors the residue found at the corresponding position in human KIF5B (L335) and in other kinesins (Fig. 3d). We then assessed the velocity and run-length of this variant and determined its two-heads-bound structure (Fig. 3e, f). Interestingly, the P364L mutation did not affect the motor's velocity (Fig. 3e, f). However, it resulted in a modest but significant reduction in run length (from a median length of 14.6 μm to 10.2 μm, Fig. 3e, f), highlighting the crucial role of the proline residue in KIF1A's superprocessivity.

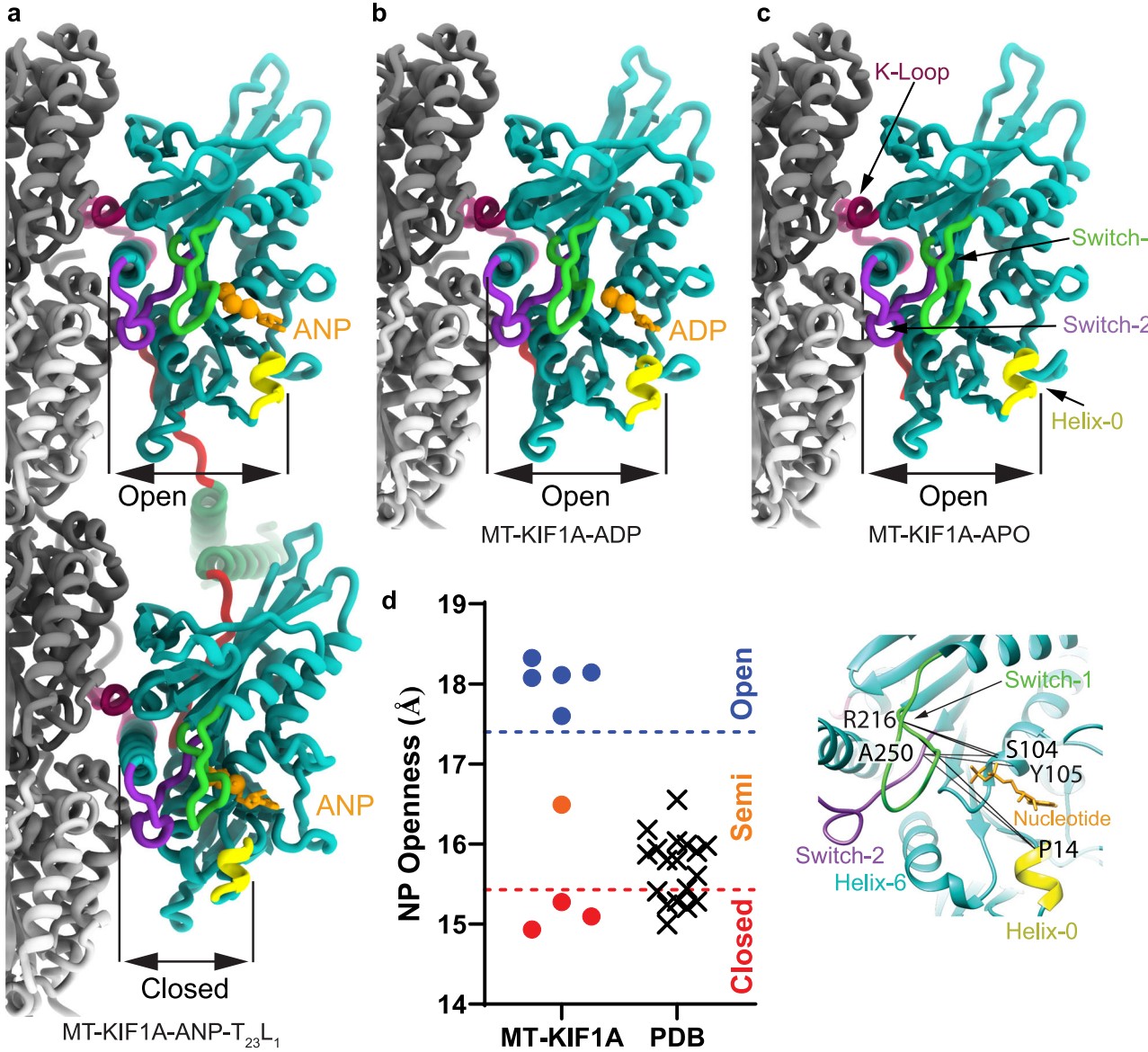

**Fig. 2 | KIF1A atomic models. a** ANP two-heads-bound configuration (MT-KIF1A-ANP-$T_{23}L_1$). **b** ADP one-head-bound configuration (MT-KIF1A-ADP). **c** Apo one-head-bound configuration (MT-KIF1A-APO). **d** Scatter plot illustrating the average of six distances between Cα carbons of selected residues (R216 and A250 to P14, to S104 and to Y105) across the KIF1A nucleotide-binding pocket (see inset). The average of these distances provides an estimate of the degree of opening of the KIF1A's nucleotide binding pocket (NP Openness). According to these measurements, the structures can be grouped into three groups, Open, Closed and Semi. The Open group consists of the MT-KIF1A motor domain structures in the ADP and Apo states, and the leading head in the two-heads-bound configuration of the ANP state (three structure classes). The only KIF1A-MT structure in the Semi group corresponds to a class of the ANP state in a single-head-bound configuration (MT-KIF1A-ANP-$T_1L_{02*}$). The cross symbols in the 'PDB' column represent KIF1A or KIF1A-MT models deposited in the PDB database, with accession codes: 1I5S, 1I6I, 1IA0, 1VFV, 1VFW, 1VFX, 1VFZ, 2HXF, 2HXH, 2ZFI, 2ZFJ, 2ZFK, 2ZFL, 2ZFM, 4UXO, 4UXP, 4UXR, 4UXS, 7EO9 and 7EOB. Structural elements surrounding the nucleotide pocket are colored as indicated in the inset. All atomic models are colored, as in Fig. 1.

The structural analysis of the P364L mutant revealed a neck-linker length similar to that of the wild-type (WT) (Fig. 3g). Nonetheless, replacing the proline likely increases conformational entropy in the substitution zone. Subtle variations in the neck-linker path and the orientation of the trailing motor domain are visible in the mutant (Fig. 3g). Such structural deviations hint at diminished interhead-tension in the P364L mutant, allowing a part of the trailing motor domain to shift slightly backward (Fig. 3g). Taken together, our functional and structural analyses suggest that the tight neck-linker connection between the two motor domains, as observed in the two-heads-bound configuration, plays a crucial role in KIF1A's enhanced processivity.

## Structure and role of the KIF1A K-loop

The specific role of KIF1A's K-loop in MT binding has been previously established[57,66,88,89], yet its interaction mechanism with MTs has remained unclear. Previous X-ray crystallography and lower-resolution cryo-EM studies have not captured the conformation of this loop and how it interacts with MTs[73–76,78,79]. However, our high-resolution structures of KIF1A-MT complexes reveal the complete polypeptide path of the K-loop (Fig. 4a). Despite the challenges in identifying the exact side-chain positions due to lower resolution, we were able to delineate the K-loop's overall trajectory. In each structure, the K-loop projects outward from the motor domain, situated opposite the nucleotide-binding site, nestled between KIF1A's helix-4 and the β-tubulin C-terminal helix (helix-12).

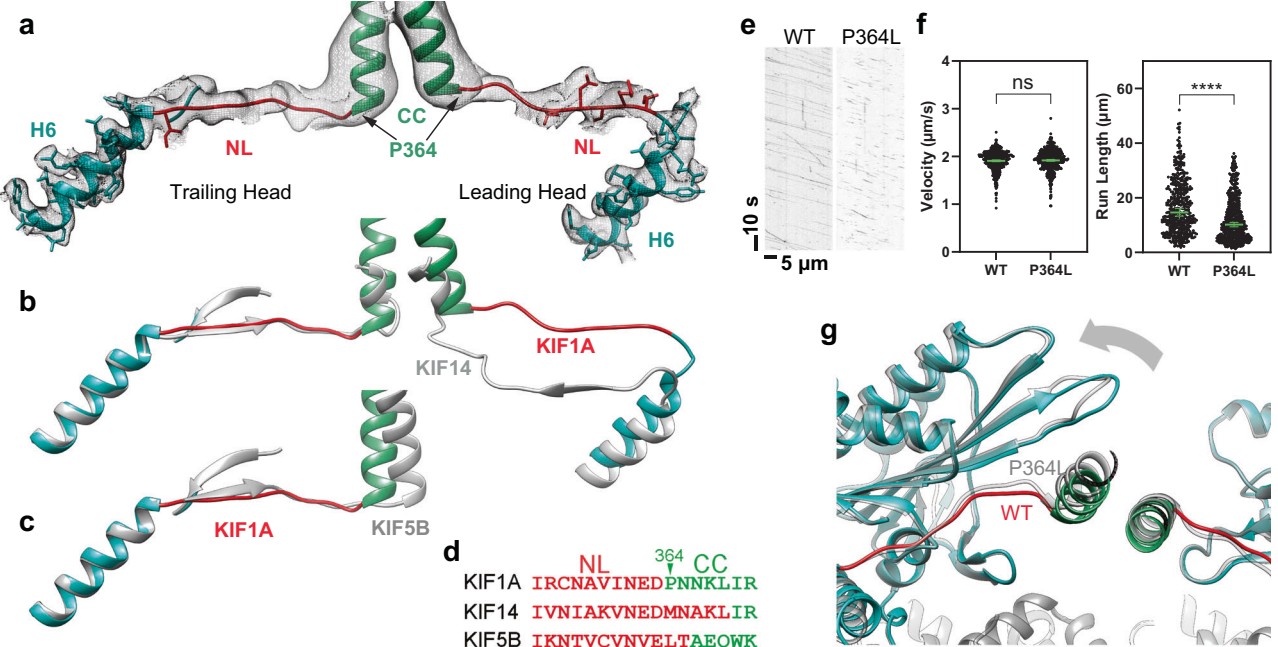

**Fig. 3 | Neck-linker structure. a** H6, neck-linker (NL) and coiled-coil (CC) region of the KIF1A ANP two-heads-bound structure (MT-KIF1A-ANP-T$_{23}$L$_1$). The structure is presented as a ribbon representation with side chain atoms as sticks and color coded as in Fig. 1. The cryo-EM map is shown as an isosurface semitransparent mesh. A composite low-pass filtered version of the map highlights the NL and CC areas, with model side chains omitted. **b** Comparison of the two-heads-bound configuration structures of KIF1A and KIF14 (gray color, PDB accession code: 6WWL). **c** Comparison of KIF1A's trailing head in the two-heads-bound configuration with KIF5B's docked NL and CC helix crystal structure (shown in gray; PDB accession code: 1MKJ). **d** Sequence alignment of NL and CC for KIF1A, KIF14 and KIF5B. Residues in the NL or the CC helix, as identified in structures (**a**) to (**c**), are colored red and green, respectively. **e** Kymograph examples showing WT and

P364L variant. **f** Velocities and run-lengths of WT and P364L variant. Green bars represent mean values for velocity or median values for run length with their respective 95% confidence interval (CI). Velocities: WT: 1.91 [1.89, 1.93] μm/s, $n = 459$; P364L: 1.92 [1.90, 1.94] μm/s $n = 514$. Run lengths: WT: 14.6 [13.3, 16.1] μm, $n = 459$; P364L: 10.2 [9.4, 11.0] μm, $n = 514$. A minimum of three experiments were performed for each construct. Statistical analysis was conducted using an unpaired two-tailed Welch's test ($P = 0.47$) for velocities and the Kolmogorov-Smirnov test for run lengths (****$P < 0.0001$). **g** Superimposed ANP two-head-bound structure models of WT (MT-KIF1A-ANP-T$_{23}$L$_1$) and P364L (MT-KIF1A$^{P364L}$-ANP-TL$_1$), aligned by their tubulin components. The WT structure is colored blue and red, while the P364L variant is represented in gray.

---

The K-loop's N-terminal lysines (K299 and K300) lie close to the β-tubulin's terminal helix (Helix-12), suggesting potential electrostatic interactions with β-tubulin Asp-427. Interestingly, other kinesin-3s, like KIF14 (Fig. 4b, c), also possess positively charged residues in this area, albeit with a less extensive loop-12[80]. This suggests that the electrostatic interactions between this loop-12 region and the MT surface may play an important role in KIF1A motility.

To probe this idea, we engineered a chimeric construct where KIF1A's K-loop was replaced with *Hs*KIF14's loop-12 (referred to as K-loop-swap or KLS mutant, see methods). In near-physiological ionic strength buffer (BRB80, see methods), this construct exhibited a slightly higher velocity compared to WT KIF1A (Fig. 4d, e). However, the run-length was significantly shorter (reducing from a median of 14.6 μm to 2.7 μm), emphasizing the unique functional role of KIF1A's K-loop in facilitating long run-lengths. This finding aligns with previous studies showing a link between prolonged MT-interaction duration and increased run lengths as the number of positively charged residues in the K-loop is increased[66,89].

Although the K-loop-swap mutant displayed a considerably shorter run-length, its median value still exceeded that of KIF5B (1.3 μm). Given that both proline 364 in the KIF1A neck-linker and the K-loop are distinct structural elements, we investigated whether combining these alterations would further reduce KIF1A's processivity to match that of KIF5B. Introducing the double mutant (P364L-Kswap) resulted in a more substantial decrease in run length (1.7 μm) than either the P364L (10.2 μm) or K-loop-swap (2.7 μm) mutants individually. However, the run length of this double mutant still marginally surpassed that of KIF5B, suggesting additional

factors in the motor domain contribute to KIF1A's superprocessivity. This is in line with previous research showing that mutations of other KIF1A-specific residues at the MT interface lead to reduced processivity[58]. In terms of translocation velocity, both the P364L and the K-loop-swap mutants either maintained or slightly increased speeds relative to WT KIF1A. Conversely, the double mutation led to a 24% decrease in velocity compared to WT, highlighting a synergistic effect between P364 in the neck-linker and the K-loop in regulating KIF1A motility.

## Interactions of the K-loop with tubulin tails

Our cryo-EM maps revealed densities at the expected location of the C-terminal tails of α- and β-tubulin, extending into the K-loop region (Fig. 5), indicative of interactions between these areas. This observation is consistent with experiments showing that the K-loop could be cross-linked with α- and β-tubulin[67]. In the cryo-EM maps, the C-terminal tails appear less pronounced and weaker compared to other areas, suggesting increased mobility. These observations provide a structural basis for the observation that removal of either the tubulin C-terminal tails or the KIF1A K-loop significantly impact MT-binding and motility[57,66,87-89].

In the case of the C-terminal tail of β-tubulin, we identified at least two concurrent positions: one aligned with the MT surface and another extending towards the K-loop (Fig. 5a, e, f). The coexisting positions may be due to high mobility or may also be a consequence of the presence of different post-translational modification or tubulin isotypes in the MT. KIF1A interactions with the β-tubulin C-terminal tail were detected in all nucleotide states (Fig. 5g).

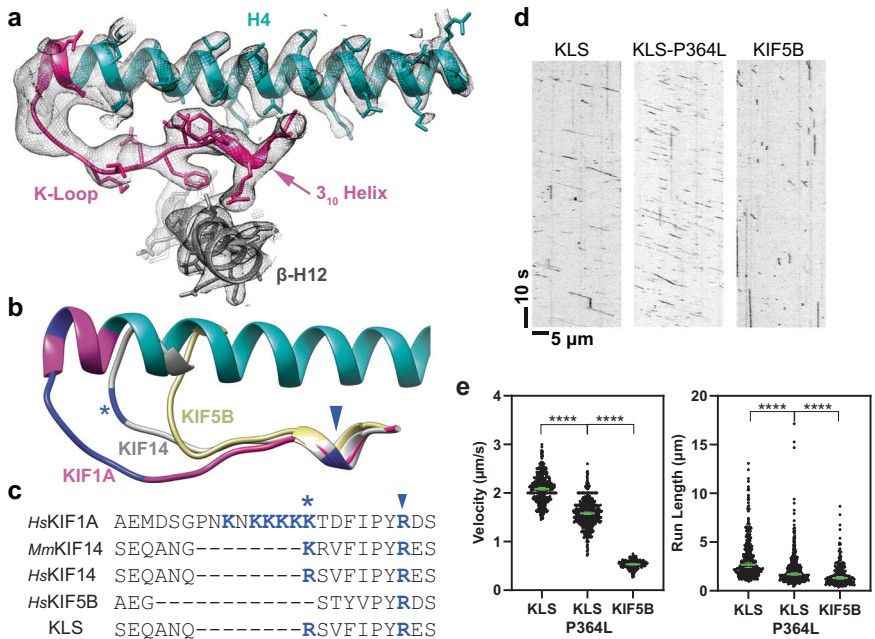

**Fig. 4 | K-loop structure. a** Area of the K-loop of the KIF1A-ADP model. The model is shown as a ribbon representation with side chain atoms as stick. Parts colored as in Fig. 1. Cryo-EM density map is represented as a semi-transparent isosurface mesh, with a low-pass filtered version of the map emphasizing the K-loop tip area (left-most section). Model side chains are omitted in this area. The $3_{10}$-helix, a component of loop-12, is a highly conserved sequence and structural motif within the kinesin superfamily. **b** Structural comparison of K-loop regions in *Hs*KIF1A, *Mm*KIF14 (PDB accession code: 6WWM) and *Hs*KIF5B (PDB accession code: 1MKJ). Positively-charged residues (K or R) within the K-loop are highlighted in dark blue. A conserved positively-charged residue in kinesin-3's loop-12 is marked with a * symbol, and a highly conserved R residue in the kinesin superfamily is indicated by an arrowhead. **c** Sequence alignments of loop-12 across different kinesins, and K-loop-swap mutant (KLS). Positively-charged residues are colored dark blue, with * and arrowhead symbols denoting the same residues marked in the structure (**b**). **d** Kymograph examples of KLS, KLS-P364L, and KIF5B. **e** Velocities and run lengths of KLS, KLS-P364L, and KIF5B. Green bars represent the mean values for velocity or median values for run length with their respective 95% CIs. Velocity: KLS: 2.09 [2.06, 2.11] μm/s $n = 398$; KLS-P364L: 1.58 [1.56, 1.61] μm/s $n = 517$; KIF5B: 0.53 [0.52, 0.54] μm/s $n = 232$. Run-lengths: KLS: 2.7 [2.4, 2.9] μm $n = 398$; KLS-P364L: 1.7 [1.6, 1.8] μm $n = 517$; KIF5B: 1.3 [1.2, 1.5] μm $n = 232$. A minimum of three experiments were performed for each construct. Statistical analysis was conducted using an unpaired two-tailed Welch's test for velocities and Kolmogorov–Smirnov test for run lengths (****$P < 0.0001$).

We also observed variations in the densities of the C-terminal tubulin tails, influenced by both the nucleotide state and the positioning of the motor domains (Fig. 5). Densities associated with the C-terminal tail of α-tubulin were only evident in the AMP-PNP state (Fig. 5a–d). Two factors may account for the difference between nucleotide states. In the two-heads-bound configuration the α-tubulin tail may help or be needed to secure MT binding of the motor-domain to the leading position. In the case of the trailing head the motor domain with its docked neck-linker, the K-loop, and a segment of the coiled-coil form a pocket of positively charged residues (Fig. 5b, c). This pocket likely attracts the α-tubulin C-tail. Moreover, the connection between the K-loop and the C-terminal tail of β-tubulin was less pronounced in the trailing motor domain compared to its leading counterpart (Fig. 5g, Supplementary Fig. 10a, b). These nucleotide-dependent variations in the interaction between the two tubulin C-terminal tails and the KIF1A motor domain provides a plausible explanation for the previously reported bias toward the MT plus-end of a monomeric KIF1A construct during one-dimensional diffusion along the MT lattice[65,66] (Supplementary Fig. 11).

In summary, our results highlight the evolutionary adaptations of KIF1A's K-loop and neck-linker. Their combined properties significantly enhance MT affinity and inter-head coordination, contributing to KIF1A's exceptional processivity.

## High-resolution structures of KIF1A-MT complexes with pathogenic P305L mutation

To advance structure-based drug development, we determined high-resolution structures of KIF1A harboring the KAND-associated P305L

mutation in complex with MTs, using the same nucleotide conditions as those for MT-bound WT KIF1A (Fig. 6a–c). Achieving sufficient MT decoration with the P305L mutant for cryo-EM imaging required a higher kinesin-to-MT ratio and a buffer with lower ionic strength compared to conditions used for the WT protein (Supplementary Table 2). Despite these adjustments, the decoration level for the mutant was lower than that of WT KIF1A (Supplementary Table 2), consistent with our previous findings of the mutant's diminished MT-binding affinity[90].

The KIF1A-P305L mutant, like its WT counterpart, exhibits classes with both one- and two-heads-bound configurations in the presence of AMP-PNP: the leading head in an open conformation and the trailing head in a closed conformation with nucleotide bound to both heads (MT-KIF1A$^{P305L}$-ANP-TL$_1$, Fig. 6a, Supplementary Fig. 12). In the ADP and Apo states, it shows a one-head-bound open conformation (Fig. 6b, c). Weak densities corresponding to α- and/or β-tubulin tails connecting to the motor domain were detected in all nucleotide states (Supplementary Fig. 10c–h), suggesting that the P305L mutation does not prevent interactions between the K-loop and the α and β-tubulin tails. Notably, in the ADP state, the P305L mutant displays a distinct map with motor domain densities appearing at much lower resolution compared to the MT densities (Fig. 6b). The lower resolution can be attributed to increased mobility, rather than poorer decoration of the mutant in the presence of ADP because the classification method separates the motor-decorated tubulins from the undecorated tubulins, and the resolution of the MT part of the map is comparable to the MT resolution of other maps presented (Supplementary Fig. 5).

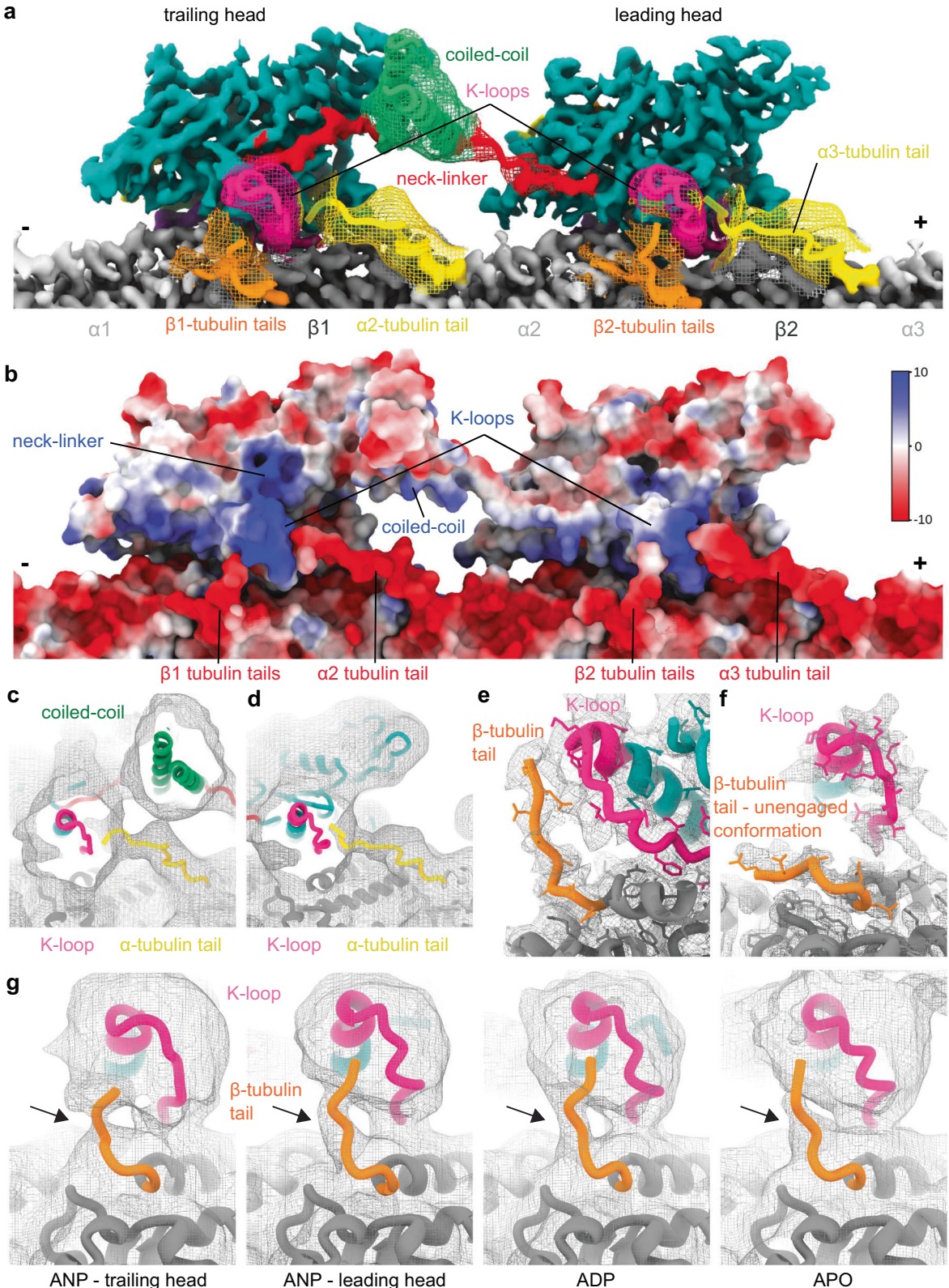

Despite the poorer resolution, densities between the tubulin tails and the KIF1A K-loop are observed (Fig. 6b), suggesting that this weakly MT-bound configuration is mediated mainly through electrostatic interactions between the K-loop and the tubulin tails. The low resolution of the kinesin part of that map prevented us from quantifying the openness of the nucleotide pocket, while the overall conformation appeared similar to that of WT KIF1A-ADP. We propose that

the P305L mutation hinders the formation of a strongly MT-bound configuration, thereby increasing the fraction of weakly attached motor domains in the MT images. Supporting this hypothesis, the P305L-ADP dataset has the lowest level of kinesin decoration (~9%) among all datasets (Supplementary Table 2).

To understand the structural impact of the P305L mutation on MT binding, we compared the mutant and WT KIF1A structures near the

**Fig. 5 | Interaction of the K-loop with the C-terminal tubulin tails. a** Composite cryo-EM map of KIF1A microtubule two-heads-bound configuration (MT-KIF1A-ANP-T$_{23}$L$_1$) interacting with three tubulin dimers. The locally filtered map is displayed as a solid color. The portions with weaker signals were low-pass filtered and displayed as meshes with the underlying model displayed as a ribbon representation. The coiled-coil and β-tubulin tails interacting with the kinesin were low-passed filtered to 6 Å, the α-tubulin tail densities were low-passed filtered to 8 Å and displayed at a low-density threshold. **b** Representation of the Coulombic electrostatic surface potential of the KIF1A two-heads-bound configuration model. Color scale in kcal·mol$^{-1}$ $e^{-1}$. **c, d** 8 Å low-passed filtered cryo-EM map of KIF1A two-heads-bound configuration (MT-KIF1A-ANP-T$_{23}$L$_1$), showing densities of α-tubulin C-terminal tails

reaching the K-loops of the trailing (**c**) or leading (**d**) head. **e** Cryo-EM map of the leading head of the P364L mutant (MT-KIF1A$^{P364L}$-ANP-TL$_1$) showing a mostly resolved β-tubulin tail conformation interacting with the K-loop. **f** Cryo-EM map of the trailing head of KIF1A two-heads-bound configuration (MT-KIF1A-ANP-T$_{23}$L$_1$) showing a portion of a β-tubulin tail conformation lying along the microtubule surface. **g** 6 Å low-passed filtered map of KIF1A near the K-loop in distinct nucleotide states and motor-domain conformations. The density associated with the β-tubulin C-terminal tail is pointed with an arrow and the threshold was adjusted for each map. Note that in the ANP state the β-tubulin tail density is better resolved in the leading head than in the trailing head.

mutated P305L residue (Fig. 6d–f). P305 is a key component of the MT-binding interface, located within the highly conserved loop-12 motif, (PYRD/E), which forms a 3$_{10}$-helix in various kinesins, including KIF1A[90]. Although previous assumptions suggested that the P305L mutation might disrupt this helix[90], our findings indicate subtler conformational shifts (Fig.6d–f). The integrity of the 3$_{10}$-helix remains intact, with most changes occurring towards the N-terminal side of the mutation. A notable divergence in the P305L mutant is the reorientation of F303 due to clash with the introduced leucine residue (Fig. 6f).

The observed structural differences between the P305L mutant and WT KIF1A suggest a potential approach for restoring the functionality of the P305L mutant by introducing a residue at position 303 that would not clash with the leucine at position 305 of the mutant. To test this hypothesis, we analyzed the motility of both a KIF1A F303V single mutant and an F303V/P305L double mutant (Fig. 6g, h). Valine was chosen due to its smaller size and hydrophobic nature, and its occurrence at this position in certain kinesin-3s, such as Unc-104. The F303V single mutant displayed reduced velocities and run lengths, along with a slight increase in landing rate (Fig. 6h and Supplementary Fig. 13), compared to WT KIF1A. Intriguingly, the double mutant exhibited an increased run length and landing rate relative to the P305L mutant alone (Fig. 6h and Supplementary Fig. 13), suggesting a partial rescue of MT-binding affinity and processivity in the double mutant.

## Discussion

KIF1A, a key member of the kinesin-3 family, is recognized for its high processivity and fast MT plus-end-directed motility. These attributes are crucial for the long-distance transport of cargoes in neurons[3–5]. Despite its importance, the underlying mechanisms of KIF1A's distinctive motility properties have remained largely undefined. Notably, most of the approximately 150 KAND mutations are within the motor domain[9], yet high-resolution structures of MT-bound KIF1A were previously unavailable and intermediates in the mechanochemical cycle such as the two-heads-MT-bound configuration had never been observed. This lack of detailed structural information, particularly concerning intermediates like the two-heads-MT-bound configuration, has hindered a deeper understanding of KIF1A's motility mechanism and limited the scope of structure-based drug discovery efforts for KAND. In this study, we have bridged this knowledge gap by determining high-resolution structures of dimeric, MT-bound KIF1A across various nucleotide states and by solving high-resolution structures of the KAND-associated P305L mutant. These structures have revealed previously unseen structural features essential for KIF1A's motility mechanism, including an open conformation of the nucleotide-binding pocket, the MT-bound conformation of KIF1A's K-loop, the dimeric structure of KIF1A in a two-heads-bound configuration, and the structural changes in the motor domain caused by the KAND P305L mutation.

In the ADP or Apo state, KIF1A is characterized by a single MT-bound motor domain, wherein the nucleotide-binding pocket adopts a markedly more open conformation than what has been documented in previous 'free', MT-unbound crystal structures of KIF1A-ADP[73,75]. This

open conformation is reminiscent of the ones observed in high-resolution structures of tubulin- and MT-bound kinesin complexes from various subfamilies[80,81,91,92]. Thus, the open conformation identified in our KIF1A-MT complexes underscores the existence of a conserved mechanism throughout the kinesin superfamily to couple MT binding with product release[72].

In previous structural studies, the K-loop of KIF1A has consistently remained unresolved[67,73,78,79], leading to a question about whether its elusiveness was due to intrinsic disorder, mobility of the loop, limitations of cryo-EM resolution, or a combination thereof. Our high-resolution structures of KIF1A-MT complexes now reveal the complete polypeptide path of the K-loop, although with a resolution that is reduced relative to other regions, suggesting higher mobility in this area. These findings imply that earlier limitations in observing the K-loop may have been related to the resolution limits of cryo-EM at the time. However, the persistent absence of a fully resolved K-loop in high-resolution of X-ray crystallography structures suggests that the loop possesses inherent disorder in the MT-unbound KIF1A. Upon MT binding, we observe that the K-loop attains a degree of structural order, becoming discernible in the context of its association with MTs. This ordering upon MT binding echoes patterns seen in other kinesins, where interaction with MTs induces structuring in parts of the motor domain that form the MT-binding interface[78,80,93,94].

Our study illuminates the structural positioning of the K-loop in KIF1A, revealing that while the bulk of the loop, including its key lysine residues, is distanced from the main body of the MT, it engages in dynamic electrostatic interactions with the C-terminal tails of both α- and β-tubulin. The K-loop effectively acts as an electrostatic bridge sandwiched between the two tubulin tails. This observation provides a structural basis for understanding the reduced run length we observed when we replaced KIF1A's K-loop with the loop-12 of the kinesin-3 KIF14, which contains a single positively-charged residue near the MT interface. This finding aligns with recent studies, highlighting the crucial role of the K-loop and its positively charged residues in facilitating KIF1A's superprocessivity[88,89].

In support of previous models for KIF1A motility[66], our structural and functional insights indicate that the K-loop's impact on KIF1A's superprocessivity primarily stems from dynamic electrostatic interactions between the K-loop and the MT. Such connections offer a structural foundation for a highly dynamic MT-bound configuration, where a single motor domain executes directionally biased one-dimensional diffusion along the MT lattice[65]. This phenomenon has been shown to rely on the K-loop and the C-terminal tubulin tails[66]. Moreover, we observe nucleotide-dependent interactions between the α- and β-tubulin C-terminal tails with the K-loop and possibly the coiled-coil domain. This provides a plausible mechanism for the reported plus-end-biased, one-dimensional diffusion of monomeric KIF1A[65] (Supplementary Fig. 11).

Our structural findings can be integrated into an updated model for KIF1A's processive motion along MTs, which aligns with our results and prior observations (Fig. 7). As in other models[83,95], it depicts the two motor domains alternating between MT-bound and unbound configurations. The model incorporates that the activity of both motor

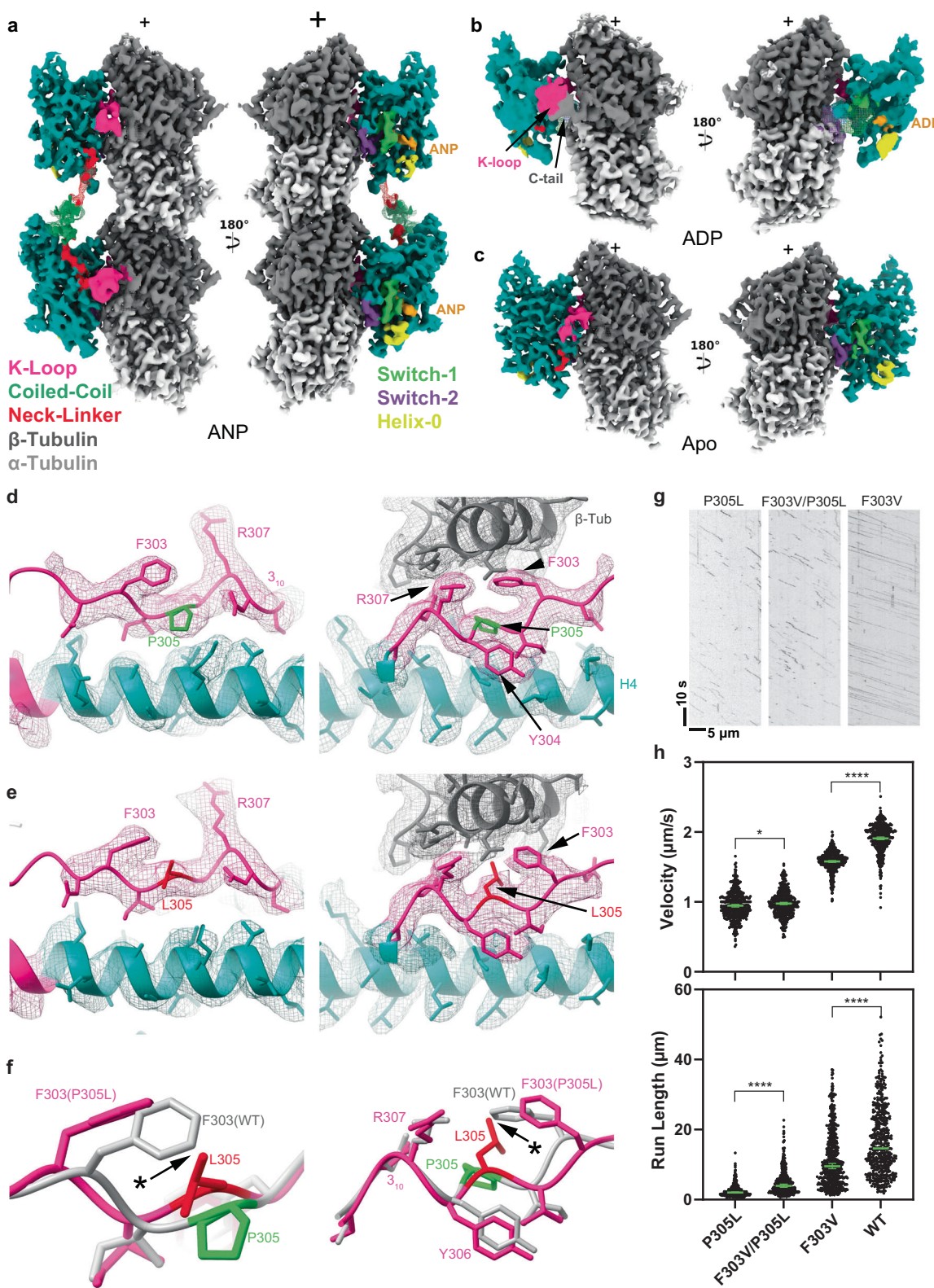

domains is coordinated to ensure that their ATPase cycles remain 'out-of-phase,' which effectively reduces the likelihood of both heads assuming simultaneously the low-affinity ADP state and preventing detachment from the MT. The KIF1A-MT structures provide evidence for at least three points of coordination between motor domains in the KIF1A mechanochemical cycle (Fig. 7 legend). In addition, in scenarios where only one head is bound or both heads transiently enter the low-affinity ADP state, the electrostatic interactions between the C-terminal tubulin tails and KIF1A's K-loop reduce the likelihood of complete detachment and may facilitate transition to a strongly MT-bound configuration, which are critical features for KIF1A's super-processive motion.

The prevailing model attributes KIF1A's superprocessivity primarily to the K-loop's interaction with tubulin[89], rather than tight

**Fig. 6 | Cryo-EM maps of microtubule-bound KIF1A$^{P305L}$ in different nucleotide states. a** MT-bound KIF1A$^{P305L}$ with ANP. The map represents the two-heads-bound configuration MT-KIF1A$^{P305L}$-ANP-TL$_1$. The coiled-coil density and part of the neck-linker from the leading head were low-pass filtered and displayed as a mesh. **b** Isosurface representation of KIF1A$^{P305L}$ in the ADP state (dataset MT-KIF1A$^{P305L}$-ADP) with the KIF1A switches area displayed as a low-pass filtered mesh. The K-loop area and C-terminal tail of β-tubulin, with connected densities, are low-passed filtered and displayed as a solid surface. **c** Isosurface representations of KIF1A$^{P305L}$ in the Apo state (dataset MT-KIF1A$^{P305L}$-APO), illustrated as in (**a**). **d** Isosurface representations of the cryo-EM map of WT KIF1A in the ANP state (MT-KIF1A-ANP-T$_{23}$L$_1$) in the neighborhood of residue P305. The surface is represented as a mesh and colored based on the underlying fitted molecular model (same color convention as in Fig. 1). The model is shown as ribbon representation with displayed side chains. The left and right panels represent two different views of the same area, the left one looking from β-tubulin, on an axis orthogonal to the microtubule axis, and the right one along the microtubule axis towards the microtubule minus end. **e** Isosurface representations of the cryo-EM map of KIF1A$^{P305L}$ mutant in the ANP state (MT-KIF1A$^{P305L}$-ANP-TL$_{012*}$), same view as in (**d**). **f** Overlay of the models displayed in (**d**) and (**e**) with the KIF1A WT model displayed in gray, the KIF1A$^{P305L}$ mutant model in pink and the residue 305 colored as in (**d**, **e**). **g** Kymograph examples of P305L, F303V/P305L and F303V. **h** Distribution of velocities and run lengths. The green bars represent the mean for velocity or median values for run length with their respective 95% confidence interval. Velocity: P305L: 0.95 [0.93, 0.96] μm/s, $n = 438$; F303V/P305L: 0.98 [0.96, 0.99] μm/s $n = 441$; F303V: 1.58 [1.57, 1.59] μm/s $n = 557$; WT: 1.91 [1.89, 1.93] μm/s, $n = 459$. The statistics were performed using an unpaired two-tailed Welch's test (****$P < 0.0001$; *$P = 0.016$). At least 3 experiments were performed for each construct, $n$ is the total number of measurements pooled from all experiments for each construct. Run lengths: P305L: 2.0 [1.9, 2.2] μm $n = 438$; F303V/P305L: 3.9 [3.5, 4.4] μm $n = 441$; F303V: 9.5 [8.8, 10.3] μm $n = 557$; WT: 14.6 [13.3, 16.1] μm $n = 459$. The statistics were performed using Kolmogorov–Smirnov test (****$P < 0.0001$).

coordination between the two motor domains. This perspective was partly informed by the assumption that KIF1A's neck-linker is longer than that of kinesin-1[89,96]. Contrary to this, our structures reveal that the KIF1A neck-linker is actually shorter than that of kinesin-1 and other kinesins, suggesting a tighter connection between the motor domains when both are bound to the MT. Furthermore, our mutagenesis studies indicate that altering KIF1A's neck-linker −by substituting the proline at the end of the neck-linker with the equivalent residue in kinesin-1− markedly reduces the motor's run length. These results collectively favor a model where KIF1A's superprocessivity is a synergistic outcome of enhanced MT binding via the K-loop and tight coordination between both motor domains.

Our model also challenges traditional views of the mechanochemical cycle by introducing a MT two-heads-bound configuration with the leading head containing ATP (Fig. 7, step 5). This contrasts with the prevailing model, which states that ATP binding to the leading head in the two-heads-bound configuration is prevented and occurring only after the trailing head detaches from the MT[77,83,95,97]. The structural basis for this model is derived from an hypothesis generated through the analysis of several crystal structures of monomeric KIF1A[77]. In this hypothesis it is postulated that the initial segment of the neck-linker (NIS, KIF1A residues A351 to I354) needs to be docked onto the motor domain for the open conformation to be stabilized. Backward pulling forces (inter-head tension) exerted on the neck-linker of the leading head would cause the NIS to undock, destabilizing the open conformation and preventing nucleotide exchange and ATP binding to the leading head[77]. However, our high-resolution structures of MT-bound KIF1A (present work) and KIF14[80] challenge several aspects of this hypothesis. They reveal an open conformation not previously observed in the KIF1A crystal structures (Fig. 1) and demonstrate that the NIS is undocked when the motor domain is in the open conformation (Supplementary Fig. 14). Our structural data instead pose an alternative model where nucleotide pocket closure, rather than ATP binding, is the step impeded by a backward-pointing neck-linker. Firstly, we have observed a two-heads-bound configuration with nucleotides bound to both heads in KIF1A, a finding also supported by the high-resolution structures of KIF14[80]. Secondly, the structural data do not support the notion that ATP binding to the leading head is prevented. In the MT two-heads bound configuration, the leading head exhibits a fully open nucleotide pocket conformation, similar to that of the MT-bound Apo and ADP states (Fig. 2, Supplementary Fig. 8).

In the two-heads-bound configuration our model predicts that ATP hydrolysis will occur first in the trailing head, as this is the one in the closed catalytic conformation. At saturating and physiological concentrations of ATP the dominant two-heads-bound configuration will have ATP bound in both heads (Fig. 7 step 5) or ATP in the leading head and the hydrolysis products ADP-Pi in the trailing head. At limiting concentrations of ATP, or if the combined rates of ATP

hydrolysis, product release, and rear head MT detachment were faster than the rate of reattachment, then during translocation, the motor would spend a greater fraction of time in the one-head-bound configuration[85,98,99].

Also contrary to previous proposals suggesting that neck-linker docking in KIF1A and kinesin-1 occurs after ATP hydrolysis rather than upon ATP binding[85,98], our data for KIF1A align with studies of other kinesins that associate neck-linker docking to ATP binding[80,100–102]. Intriguingly, in the ATP-bound state of KIF1A, we observed a significant proportion of molecules in a one-head-bound conformation with a docked neck-linker. This observation raises the possibility that ATP hydrolysis might facilitate the binding of the tethered head to the leading position, rather than being a prerequisite for neck-linker docking. However, our structural data also indicates that binding of the leading head is necessary for a more complete closure of the nucleotide-binding pocket of the trailing head (~50% of the motors in the AMP-PNP one-head-bound configuration where in a semi-closed conformation, Fig. 2d, Supplementary Figs. 3f and 4). Assuming that the closed conformation is the catalytic one[102], this would suggest that ATP hydrolysis is more favorable after the two-heads-bound configuration occurs.

Our data also elucidate the impact of the KAND mutation P305L on MT-binding (Fig. 7 steps 1–2). Unexpectedly, the P305L mutation primarily influences the orientation of F303, rather than the adjacent highly conserved 3$_{10}$-helix region of loop-12, which was previously shown to be critical for KIF1A-MT binding[90,103]. Interestingly, in the highest-scoring AlphaFold2[104,105] (AF2) structure predictions for the MT-unbound P305L mutant, F303 is positioned in a manner that would clash with the β-tubulin helix H12, rendering it incompatible with MT binding (Supplementary Fig. 15b, d, f). On the other hand, the five best AF2 models of WT KIF1A more closely resemble the MT-bound structures in this area (Supplementary Fig. 15a, c, e). These predictions, along with the conformational changes observed in the MT-bound structures, indicate that the P305L mutation induces loop-12 conformations that are suboptimal for MT binding, thereby shifting the equilibrium towards the MT-unbound (or weakly-bound) states, as observed here. Furthermore, our experiments reveal that the introduction of the F303V mutation alone reduced processivity, emphasizing the vital role of this residue in KIF1A's motility. A similar modification of the corresponding residue in kinesin-1's loop-12 also resulted in a reduced MT affinity[103]. Notably, combining the P305L and F303V mutations partially restored processivity, suggesting that this region of loop-12 is a viable target for future structure-guided design strategies.

In summary, our findings support a revised model for the kinesin mechanochemical cycle and KIF1A superprocessivity. This model posits that KIF1A superprocessivity arises from enhanced K-loop-mediated interactions with MTs and coordination between the two

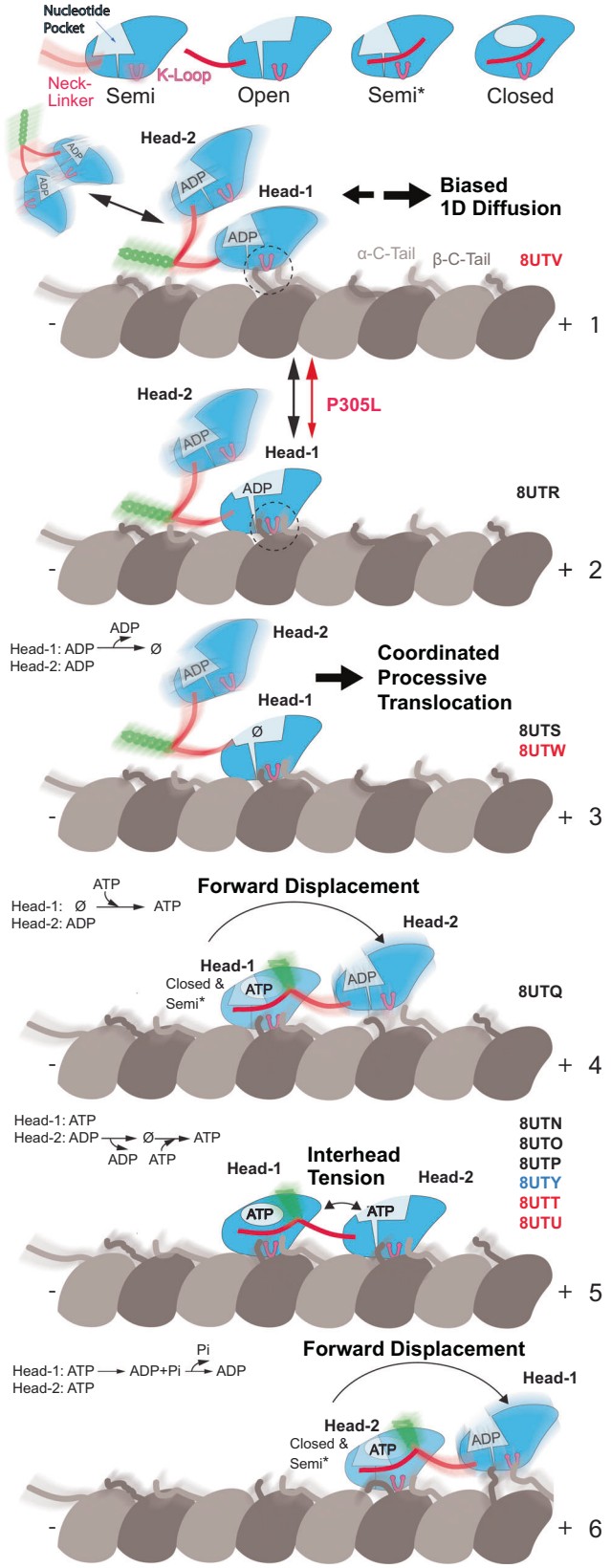

**Fig. 7 | A model for KIF1A's mechanochemical cycle.** The upper row displays the four primary conformations of the KIF1A motor domain, named based on the degree of openness of the nucleotide-binding pocket. Two types of KIF1A-MT interactions or binding modes can be inferred from the structures, a strong binding mode mediated by stereospecific contacts between the MT and the whole KIF1A-MT interface, and a weak-binding mode mediated by flexible electrostatic interactions between the tubulin C-terminal tails, the KIF1A K-loop and possibly the neck-linker and the start of the coiled-coil domain. Regions of the structures with high mobility are illustrated with a blurred depiction. The PDB accession codes of the structure(s) representing each model step are displayed to the right, with different font colors indicating the KIF1A construct used, black for WT, red for the P305L mutant, and blue for the P364L mutant. The cycle starts with a KIF1A dimer in the ADP state interacting with the MT in a weakly bound mode (state 1). In this state, KIF1A could engage in biased one-dimensional diffusion (Supp. Fig. 11) or in directed motion, where the two motor heads coordinate alternating conformations, nucleotide states, and MT binding modes to move in a hand-over-hand manner in the MT plus-end direction (steps 2 to 6). The P305L mutation alters the structure of the KIF1A MT interface, hindering the transition to the strongly bound configuration (from step 1 to step 2). Several points of the coordination between the two heads can be derived from the structures: In steps 2 and 3 one head is prevented from binding to the MT until the MT-bound head binds ATP (steps 4 and 5). In step 4, the MT-bound head with ATP (head-1) has a reduced probability to reach the fully closed conformation until the partner motor domain binds strongly to the MT in the forward position (step 5). In the two-heads MT-bound configuration (step 5), inter-head tension and the differently oriented neck-linkers maintain the two heads in distinct conformations. In this configuration, nucleotide pocket closure and ATP hydrolysis in the leading head is paused until the trailing head detaches allowing the neck-linker of the leading head to dock. After ATP hydrolysis, product release and detachment of the rear head the neck-linker of the leading head docks moving the detached head forward (step 6).

of KIF1A are pivotal for developing targeted treatments, especially for KAND and related neurological disorders.

## Methods

### Generation of plasmids for KIF1A constructs

A plasmid for a previous published KIF1A construct[9] (KIF1A (*Homo sapiens*, aa 1-393)-leucine zipper-SNAPf-EGFP-6His) was used as the template for all constructs in this study. For proteins used in the cryo-EM studies, the SNAPf-EGFP-6His tag was replaced with a strep-II tag (IBA Lifesciences GmbH) using Q5 mutagenesis (New England Biolabs Inc., #E0554S). Mutations within KIF1A were generated using Q5 mutagenesis. The K-loop swap mutant has EQANQRSV instead of the amino acid sequence of EMDSGPNKNKKKKKTD (KIF1A residues 287-302). All plasmids were confirmed by sequencing.

### Protein expression in *E. coli*

KIF1A expression was performed as following[9,52]. Each plasmid was transformed into BL21-CodonPlus(DE3)-RIPL competent cells (Agilent Technologies, #230280). A single colony was picked and inoculated in 1 mL of terrific broth (TB) (protocol adopted from Cold Spring Harbor protocol, https://doi.org/10.1101/pdb.rec8620) with 50 μg/mL carbenicillin and 50 μg/mL chloramphenicol. The 1-mL culture was shaken at 37 °C overnight, and then inoculated into 400 mL of TB (or 1–2 L for cryo-EM studies) with 2 μg/mL carbenicillin and 2 μg/mL chloramphenicol. The culture was shaken at 37 °C for 5 hours and then cooled on ice for 1 hour. IPTG was then added to the culture to a final concentration of 0.1 mM to induce expression. Afterwards, the culture was shaken at 16 °C overnight. The cells were harvested by centrifugation at 3,000 × g for 10 minutes at 4 °C. The supernatant was discarded, and 1.25 mL of B-PER™ Complete Bacterial Protein Extraction Reagent (ThermoFisher Scientific, #89821) per 100 mL culture with 2 mM MgCl$_2$, 1 mM EGTA, 1 mM DTT, 0.1 mM ATP, and 2 mM PMSF was added to the cell pellet. The cells were fully resuspended, and flash frozen in liquid nitrogen. If the purification was not done on the same day, the frozen cells were stored at −80 °C.

motor domains. Not only does this revised model deepen our understanding of KIF1A's superprocessivity, but also sheds light on how mutations like P305L impact its function. Our structural insights into these mutations, especially their unexpected effects on MT binding, highlight potential areas for therapeutic interventions. These advancements in understanding the structural and functional aspects

## Protein purification

To purify the protein, the frozen cell pellet was thawed at 37 °C. The solution was nutated at room temperature for 20 min and then dounced for 10 strokes on ice to lyse the cells. Unless specified, the following procedures were done at 4 °C. The cell lysate was cleared by centrifugation at 80,000 rpm (260,000 × g, k-factor = 28) for 10 minutes in an TLA-110 rotor using a Beckman Tabletop Ultra-centrifuge Unit. The supernatant was flown through 500 μL of Roche cOmplete™ His-Tag purification resin (Millipore Sigma, #5893682001) for His-tag tagged proteins, or 2 mL of Strep-Tactin® Sepharose® resin (IBA Lifesciences GmbH, #2-1201-002) for strep-II tagged proteins. The resin was washed with wash buffer (WB) (for His-tagged protein: 50 mM HEPES, 300 mM KCl, 2 mM MgCl$_2$, 1 mM EGTA, 1 mM DTT, 1 mM PMSF, 0.1 mM ATP, 0.1% Pluronic F-127 (w/v), 10% glycerol, pH 7.2; for strep-II tagged protein, Pluronic F-127 and glycerol were omitted). For proteins with a SNAPf-tag, the resin was mixed with 10 μM SNAP-Cell® TMR-Star (New England Biolabs Inc., #S9105S) at room temperature for 10 minutes to label the SNAPf-tag. The resin was further washed with WB, and then eluted with elution buffer (EB) (for His-tagged protein: 50 mM HEPES, 150 mM KCl, 150 mM imidazole, 2 mM MgCl$_2$, 1 mM EGTA, 1 mM DTT, 1 mM PMSF, 0.1 mM ATP, 0.1% Pluronic F-127 (w/v), 10% glycerol, pH 7.2; for strep-II tagged protein: 80 mM PIPES, 2 mM MgCl$_2$, 1 mM EGTA, 1 mM DTT, 0.1 mM ATP, 5 mM desthiobiotin). The Ni-NTA elute was flash frozen and stored at −80 °C. The Strep-Tactin elute was concentrated using an Amicon Ultra-0.5 mL Centrifugal Filter Unit (30-kDa MWCO) (Millipore Sigma, #UFC503024). Storage buffer (SB) (80 mM PIPES, 2 mM MgCl$_2$, 80% sucrose (w/v)) was added to the protein solution to have a final 20% sucrose (w/v) concentration, and the protein solution was flash frozen and stored at −80 °C. The purity of the proteins was confirmed on polyacrylamide gels (Supplementary Fig. 1).

## Microtubule-binding and -release assay

An MT-binding and -release (MTBR) assay was performed to remove inactive motors for single-molecule TIRF assay. 50 μL of eluted protein was buffer-exchanged into a low salt buffer (30 mM HEPES, 50 mM KCl, 2 mM MgCl$_2$, 1 mM EGTA, 1 mM DTT, 1 mM AMP-PNP, 10 μM taxol, 0.1% Pluronic F-127 (w/v), and 10% glycerol) using 0.5-mL Zeba™ spin desalting column (7-kDa MWCO) (ThermoFisher Scientific, #89882). The solution was warmed to room temperature and 5 μL of 5 mg/mL taxol-stabilized MTs was added. The solution was well mixed and incubated at room temperature for 2 minutes to allow motors to bind to the MTs and then spun through a 100 μL glycerol cushion (80 mM PIPES, 2 mM MgCl$_2$, 1 mM EGTA, 1 mM DTT, 10 μM taxol, and 60% glycerol, pH 6.8) by centrifugation at 45,000 rpm (80,000 × g, k-factor = 33) for 10 minutes at room temperature in TLA-100 rotor using a Beckman Tabletop Ultracentrifuge Unit. Next, the supernatant was removed and the pellet was resuspended in 50 μL high salt release buffer (30 mM HEPES, 300 mM KCl, 2 mM MgCl$_2$, 1 mM EGTA, 1 mM DTT, 10 μM taxol, 3 mM ATP, 0.1% Pluronic F-127 (w/v), and 10% glycerol). The MTs were then removed by centrifugation at 40,000 rpm (60,000 × g, k-factor = 41) for 5 minutes at room temperature. Finally, the supernatant containing the active motors was aliquoted, flash frozen in liquid nitrogen, and stored at −80 °C.

## Single-molecule TIRF motility assay

MTBR fractions were used for the single-molecule TIRF assay and the dilutions were adjusted to an appropriate density of motors on MTs. The assay was performed as follows[106]. A flow chamber was assembled with a glass slide (Fisher #12-550-123), an ethanol-cleaned coverslip (Zeiss #474030-9000-000), and two stripes of parafilm. All the following incubation was done at room temperature. 10 μl of 0.5 mg/ml BSA-biotin was flown into the chamber with 10 min incubation. The chamber was then washed with 2 × 20 μl blocking buffer (80 mM PIPES, 2 mM MgCl$_2$, 1 mM EGTA, 10 μM taxol, 1% Pluronic F-127 (w/v), pH 6.8)

and incubated for 10 min to block the surface. Afterwards, 10 μl of 0.25 mg/ml streptavidin was introduced into the chamber with 10 min incubation. The chamber was washed with 2 × 20 μl blocking buffer, and 10 μl of 0.02 mg/ml Cy5- and biotin-labeled MTs were flown into the chamber with 1 min incubation. The chamber was washed with 2 × 20 μl blocking buffer. The MTBR motor was diluted in motility buffer (80 mM PIPES, 2 mM MgCl$_2$, 1 mM EGTA, 1 mM DTT, 10 μM taxol, 0.5% Pluronic F-127 (w/v), 2 mM ATP, 5 mg/mL BSA, 1 mg/mL α-casein, gloxy oxygen scavenging system, and 10% glycerol, pH 6.8) in an appropriated dilution, and the solution was introduced into the chamber. The chamber was sealed with vacuum grease. Images were acquired with 200 ms per frame (total 600 frames per movie), and then analyzed via a custom-written MATLAB software. Kymographs were generated using ImageJ2 (version 2.9.0). Statistical analysis was performed and graphs were generated using GraphPad Prism (version 9.5.0).

## Cryo-EM

**Preparation of microtubules.** Microtubules (MTs) were prepared from porcine brain tubulin (Cytoskeleton, Inc. CO). Tubulin lyophilized pellets were resuspended in BRB80 (80 mM K-PIPES, 1 mM MgCl$_2$, 1 mM EGTA, pH 6.8) to 5 mg/mL and spun at 313,000 × g before polymerization to eliminate aggregates. MT polymerization was done in conditions to enrich the number of MTs with 15 protofilaments[107] as follows. The clarified resuspended tubulin solution was supplemented with 2 mM GTP, 4 mM MgCl$_2$, 12%(v/v) DMSO and incubated 40 min at 37 °C. An aliquot of stock Paclitaxel (Taxol®) solution (2 mM in DMSO) was added for a final paclitaxel concentration of 250 μM and incubated for another 40 min at 37 °C. The MTs were then spun at 15,500 × g, 25 °C and the pellet resuspended in BRB80 with 20 μM paclitaxel.

**Cryo-EM of KIF1A-MT complexes.** Four μL of 6 μM MT solution in BRB80 plus 20 μM paclitaxel were layered onto (UltrAuFoil R1.2/1.3 300 mesh) plasma cleaned just before use (Gatan Solarus plasma cleaner, at 15 W for 6 seconds in a 75% argon/25% oxygen atmosphere), the MTs were incubated 1 minute at room temperature and then the excess liquid removed from the grid using a Whatman #1 paper. Four μL of a solution of KIF1A WT in BRB80 supplemented with 20 μM paclitaxel (kinesin concentrations given in Supplementary Table 2) and either (1) 4 mM AMP-PNP, (2) 4 mM ADP, or (3) $5 \times 10^{-3}$ units per μL apyrase (APO condition) were then applied onto the EM grid and incubated for 1 minute at room temperature. The grid was mounted into a Vitrobot apparatus (FEI-ThermoFisher MA), incubated 1 min at room temperature and plunge-frozen into liquid ethane (Vitrobot settings: 100% humidity, 3 seconds blotting with Whatman #1 paper and 0 mm offset). Grids were stored in liquid nitrogen. For the P305L mutant, the kinesin solution was prepared as the solution for WT KIF1A with the difference that instead of BRB80 we used BRB36 (36 mM K-PIPES, 1 mM MgCl$_2$, 1 mM EGTA, pH 6.8) (Supplementary Table 2).

**Cryo-EM data collection.** Data were collected at 300 kV on Titan Krios microscopes equipped with a K3 summit detector. The acquisition was controlled using Leginon[108] with the image-shift protocol and partial correction for coma induced by beam tilt[109]. The pixel sizes, defocus ranges and cumulated dose are given in Supplementary Table 1.

**Processing of the cryo-EM datasets of MT-kinesin complexes.** Image processing was done as follows[80]. Movie frames were aligned with Relion generating both dose-weighted and non-dose-weighted sums. Contrast transfer function (CTF) parameters per micrographs were estimated with Gctf[110].

Helical reconstruction on 15R MTs was performed using a helical-single-particle 3D analysis workflow in Frealign[80,92,111]. Briefly: microtubule filament images were manually selected and then subjected to supervised classification to select 15R protofilament microtubules.

Particle image boxes were extracted from the selected microtubules at 8 nm intervals along the microtubule axis and each microtubule was assigned to a distinct half-set. A preliminary helically averaged 3D map was first calculated with Spider[112]. The resulting 3D map was then used as the reference model to produce a refined helical averaged 3D map with Frealign[111] with $664 \times 664$ pixel box sizes. Per-particle CTF refinement was performed with FrealignX[113].

To select for tubulins bound to kinesin motors and to improve the resolution of the kinesin-tubulin complexes, the procedure HSARC[80] was used. The procedure follows these steps:

(1) Relion helical refinement. The two independent Frealign helical refined half datasets were subjected to a single helical refinement in Relion 3.1[114] using as priors the Euler angle values determined in the helical-single-particle 3D reconstruction (initial resolution: 8 Å, sigma on Euler angles sigma_ang: 1.5, no helical parameter search).

(2) Asymmetric refinement with partial signal subtraction. Atomic models of kinesin-tubulin complexes derived from our recent work on KIF14[80] were used to generate a soft mask $mask_{full}$ using EMAN pdb2mrc and relion_mask_create (low-pass filtration: 30 Å, initial threshold: 0.05, extension: 14 pixels, soft edge: 6 Å). For the one-head-bound only datasets (MT-KIF1A-ADP, MT-KIF1A-APO, MT-KIF1A$^{P305L}$-ADP and MT-KIF1A$^{P305L}$-APO), $mask_{full}$ was generated with a model containing 1 kinesin motor bound to 1 tubulin dimer and two longitudinally flanking tubulin subunits. For the datasets containing two-heads-bound configurations (MT-KIF1A-ANP, MT-KIF1A$^{P364L}$-ANP and MT-KIF1A$^{P305L}$-ANP), $mask_{full}$ was generated from a kinesin dimer model bound to two tubulin dimers.

The helical dataset alignment file was symmetry expanded using the 15 R MT symmetry of the dataset. Partial signal subtraction was then performed using $mask_{full}$ to retain the signal within that mask. During this procedure, images were re-centered on the projections of 3D coordinates of the center of mass of $mask_{full}$ ($C_M$) using a 416 pixels box size. The partially signal subtracted dataset was then used in a Relion 3D refinement procedure using as priors the Euler angle values determined form the Relion helical refinement and the symmetry expansion procedure (initial resolution: 8 Å, sigma_ang: 2 or 5, healpix_order and auto_local_healpix_order set to 5). The CTF of each particle was corrected to account for their different position along the optical axis.

(3) 3D classification of the kinesin signal. A mask $mask_{kinesin}$ was generated like in step (2) but using only the kinesin coordinates (a single kinesin head for the datasets MT-KIF1A-ADP, MT-KIF1A-APO, MT-KIF1A$^{P305L}$-ADP and MT-KIF1A$^{P305L}$-APO; a two-heads-bound kinesin dimer for the dataset MT-KIF1A-ANP, MT-KIF1A$^{P364L}$-ANP and MT-KIF1A$^{P305L}$-ANP). A second partial signal subtraction procedure identical to first one in step (2) but using $mask_{kinesin}$ instead of $mask_{full}$, with particles re-centered on the projections of $C_M$ was performed to subtract all but the kinesin signal. The images obtained were resampled to 3.5 Å/pixel in 100-pixel boxes and the 3D refinement from step 2 was used to update the Euler angles and shifts of all particles.

For the datasets MT-KIF1A-ADP, MT-KIF1A-APO, MT-KIF1A$^{P305L}$-ADP and MT-KIF1A$^{P305L}$-APO with a single-head-bound configuration, a 3D focused classification without images alignment and using a mask for the kinesin generated like $mask_{kinesin}$ (low-pass filtration: 30 Å, initial threshold: 0.9, extension: 1 pixel, soft edge: 3 pixels) was then performed on the resampled dataset (8 classes, tau2_fudge: 4, padding: 2, iterations: 175). The class(es) showing a kinesin were selected and merged for the next step. For the MT-KIF1A$^{P305L}$-ADP which had very low decoration (~9 %, Supplementary Table 2), the previous classification led to a single decorated class (14% of the dataset) with a weak kinesin

signal. Particles from this class were further classified in a second 3D classification (4 classes, tau2_fudge: 4, padding: 2, iterations: 175) and the main decorated class (68%) with a recognizable kinesin was selected while the others (with resolution lower than 15 Å) were not.

For the MT-KIF1A-ANP, MT-KIF1A$^{P364L}$-ANP and MT-KIF1A$^{P305L}$-ANP datasets, 3D classifications in 8 classes performed as described above on the partially subtracted data with 2 kinesin motor domains left revealed that both one-head- and two-heads-bound configurations were present but not fully separated. The following hierarchical 3D-classification was used instead (Supplementary Figs. 3, 4, and 12). Two different masks were generated (Supplementary Fig. 3a–c): one covering the kinesin site closest to the (−) end of the MT called $mask_T$, and the other covering the kinesin site closest to the (+) end of the MT, called $mask_L$. These two masks cover respectively the trailing head and leading head of a two-heads-bound kinesin dimer bound to these two sites (sites T and L respectively, Supplementary Fig. 3a–c). Each of these two masks was generated from an atomic model corresponding to the overlay of a trailing head and a leading head of a two-heads-bound kinesin dimer with its associated coiled-coil, so that the focused classification includes the signal of the coiled-coil independent of the kinesin registration (Supplementary Fig. 3d, e). These models were converted to maps with EMAN pdb2mrc and the masks were generated with relion_mask_create (low-pass filtration: 30 Å, initial threshold: 0.9-1.4, extension: 1 pixel, soft edge: 3 pixels). First a 3D classification was performed focusing the classification on $mask_T$ (8 classes, tau2_fudge: 6, padding: 2, iterations: 175, Supplementary Fig. 3f). Then a second focused classification using $mask_L$ (4 classes, padding: 2, iterations: 175) was performed on the classes obtained in the previous classification step and selecting only the classes that were potentially trailing heads of a two-heads-bound configuration (Supplementary Fig. 3f). This excluded classes that contained clear neck-linker and coiled-coil cryo-EM densities toward the (−) end of the motor-domain density (i.e. a leading head), the classes that show no kinesin density and the classes having an unrecognizable signal. Each of the main classes used was named as indicated in Supplementary Fig. 3f. The decoration and propensity of the states identified resulting from these classifications are given in Supplementary Table 2. All datasets produced at least one class where the kinesin motor densities were well-resolved.

(4) 3D reconstructions with original images (not signal subtracted). To avoid potential artifacts introduced by the signal subtraction procedure, final 3D reconstructions of each half dataset were obtained using relion_reconstruct on the original image particles extracted from the micrographs without signal subtraction. To increase the resolution on site T for the datasets MT-KIF1A-ANP and MT-KIF1A$^{P305L}$-ANP, some reconstructions were generated by merging the data from different classes obtained in the classification on site T or L. In this case, the names of the classes are concatenated, for example, MT-KIF1A-ANP-$T_{23}L_1$ was generated by merging the data from MT-KIF1A-ANP-$T_2L_1$ and MT-KIF1A-ANP-$T_3L_1$.

(5) To obtain a final locally filtered and locally sharpened map for the states listed in Supplementary Table 1 (MT-KIF1A-ANP-$T_{23}L_1$, MT-KIF1A-ANP-$T_2L_1$, MT-KIF1A-ANP-$T_3L_1$, MT-KIF1A-ANP-$T_1L_{02*}$, MT-KIF1A-ADP, MT-KIF1A-APO, MT-KIF1A$^{P364L}$-ANP, MT-KIF1A$^{P305L}$-ANP-$TL_1$, MT-KIF1A$^{P305L}$-ANP-$TL_{012*}$, MT-KIF1A$^{P305L}$-ADP and MT-KIF1A$^{P305L}$-APO), post-processing of the pair of unfiltered and unsharpened half maps was performed as follows. One of the two unfiltered half-map was low-pass-filtered to 15 Å and the minimal threshold value that does not show noise around the MT fragment was used to generate a soft mask with relion_mask_create (low-pass filtration: 15 Å). This soft mask was used in blocres[115] on

12-pixel size boxes to obtain initial local resolution estimates. The merged map was locally filtered by blocfilt[115] using blocres local resolution estimates and then used for local low-pass filtration and local sharpening in localdeblur[116] with resolution search up to 25 Å. The localdeblur program converged to a filtration appropriate for the tubulin part of the map but over-sharpened for the kinesin part. The maps at every localdeblur cycle were saved and the map with better filtration for the kinesin part area was selected with the aim of improving the resolution of the kinesin loops.

The final reconstructions of MT-KIF1A-ANP-$T_1L_1$, MT-KIF1A-ANP-$T_1L_0$, MT-KIF1A-ANP-$T_2L_0$, MT-KIF1A-ANP-$T_3L_0$, MT-KIF1A-ANP-$T_1L_*$, MT-KIF1A-ANP-$T_2L_*$, MT-KIF1A-ANP-$T_3L_*$ (Supplementary Fig. 4) were low-passed filtered at 4.0 Å and sharpened with a b-factor of −40 A$^2$.

**Cryo-EM resolution estimation.** The final resolutions for each cryo-EM class average reconstruction listed in Supplementary Table 1 (MT-KIF1A-ANP-$T_{23}L_1$, MT-KIF1A-ANP-$T_2L_1$, MT-KIF1A-ANP-$T_3L_1$, MT-KIF1A-ANP-$T_1L_{02*}$, MT-KIF1A-ADP, MT-KIF1A-APO, MT-KIF1A$^{P364L}$-ANP MT-KIF1A$^{P305L}$-ANP-TL$_1$, MT-KIF1A$^{P305L}$-ANP-TL$_{012*}$, MT-KIF1A$^{P305L}$-ADP and MT-KIF1A$^{P305L}$-APO) were estimated from FSC curves generated with Relion 3.1 post-process (FSC$_{0.143}$ criteria, Supplementary Fig. 5). To estimate the overall resolution, these curves were computed from the two independently refined half maps (gold standard) using soft masks that isolate a single asymmetric unit containing a kinesin and a tubulin dimer. The soft masks were created with Relion 3.1 relion_mask_create (low-pass filtration: 15-20Å, soft edge: 5-8 pixels) applied on the correctly positioned EMAN pdb2mrc density map generated with the coordinates of the respective refined atomic models. FSC curves for the tubulin or kinesin parts of the maps were generated similarly using the corresponding subset of the PDB model to mask only a kinesin or a tubulin dimer (Supplementary Fig. 5, Supplementary Table 1).

The final cryo-EM maps together with the corresponding half maps, the masks used for resolution estimation, the masks used in the partial signal subtraction for the MT datasets, the low-passed filter maps used in Fig. 5 and Supplementary Figs. 9, 10, and the FSC curves (Supplementary Fig. 5) are deposited in the Electron Microscopy Data Bank (Supplementary Table 1).

**Model building**
Atomic models of the cryo-EM density maps were built as follows. First, atomic models for each protein chains were generated from their amino-acid sequence by homology modeling using Modeller[117]. Second, the protein chains were manually placed into the cryo-EM densities and fitted as rigid bodies using UCSF-Chimera[118]. Third, the models were flexibly fitted into the density maps using Rosetta for cryo-EM relax protocols[119,120] and the models with the best scores (best match to the cryo-EM density and best molprobity scores) were selected. Fourth, the Rosetta-refined models were further refined against the cryo-EM density maps using Phenix real space refinement tools[121]. Fifth, the Phenix-refined models were edited using Coot[122]. Several iterations of Phenix real space refinement and Coot editing were performed to reach the final atomic models.

Atomic models and cryo-EM map figures were prepared with UCSF-Chimera[118] or USCF ChimeraX[123] and R[124].

**Nucleotide binding pocket openness**
Nucleotide-binding pocket (NP) openness values for each atomic model were calculated as the average of six distances between KIF1A Cα carbons of highly conserved residues (R216 and A250 to P14, to S104 and to Y105) across the kinesin nucleotide-binding pocket[82]. Atom distances were calculated from the atomic models using UCSF-Chimera[118]. The precision of the NP openness values (expressed as

mean ± standard deviation (SD)) was estimated by regenerating multiple atomic models[125]. For each atomic model eighty models were regenerated by flexibly fitting the model into the corresponding cryo-EM map using Rosetta for cryo-EM relax protocols[119,120,125]. Mean and SD were calculated for each model from six NP openness values: one from the model itself and the others from the five best-scored Rosetta-regenerated models.

**Reporting summary**
Further information on research design is available in the Nature Portfolio Reporting Summary linked to this article.

## Data availability
The data that support this study are available from the corresponding authors upon request. The structural models generated in this study have been deposited in the Protein Data Bank (PDB) database under accession codes: 8UTN, 8UTO, 8UTP, 8UTQ, 8UTR, 8UTS, 8UTT, 8UTU, 8UTV, 8UTW and 8UTY. Corresponding cryo-EM maps, half maps, FSC curves and masks used have been deposited in the Electron Microscopy Data Base (EMDB) with accession codes: EMD-42543, EMD-42544, EMD-42545, EMD-42546, EMD-42547, EMD-42548, EMD-42549, EMD-42550, EMD-42551, EMD-42552 and EMD-42553. Files containing the data plotted in the main and Supplementary Figures and Tables are provided in the Source_Data.zip zipped folder. The following previously published structural models were referenced in this work: 1I5S, 1I6I, 1IAO, 1MKJ, 1VFV, 1VFW, 1VFX, 1VFZ, 2HXF, 2HXH, 2ZFI, 2ZFJ, 2ZFK, 2ZFL, 2ZFM, 4UXO, 4UXP, 4UXR, 4UXS, 6WWL, 6WWM, 7EO9 and 7EOB. Source data are provided with this paper.

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

## Acknowledgements

This work was supported by National Institutes of Health Grant R01GM113164 (H.S.), R01GM147332 (A.G. and H.S.), R01GM098469 (A.G.), and R01NS114636 (A.G.). Cryo-EM data collection was performed at the Simons Electron Microscopy Center and National Resource for Automated Molecular Microscopy located at the New York Structural Biology Center, supported by grants from the Simons Foundation (SF349247), NYSTAR, and the NIH National Institute of General Medical Sciences (GM103310) with additional support from Agouron Institute (F00316) and NIH (OD019994).

## Author contributions

L.R. generated, expressed, and purified the constructs for the cryo-EM and single-molecule studies. L.R. performed single-molecule experiments, collected, and analyzed the data, and interpreted the results. M.P.M.H.B. and A.B.A. assembled kinesin-MT complexes and made cryo-EM grid samples; A.B.A. performed sample screening and optimization for cryo-EM imaging and performed MT selection; M.P.M.H.B., A.B.A. and H.S. designed the cryo-EM experiments; M.P.M.H.B. and A.B.A. performed cryo-EM data collection; M.P.M.H.B. designed and performed the cryo-EM data processing of kinesin-MT complexes and interpreted the results; M.P.M.H.B. and H.S. built and refined the atomic models, and interpreted the structures; A.G. and H.S. conceived and coordinated the project and interpreted the results; M.P.M.H.B., L.R, A.B.A., A.G. and H.S. wrote the manuscript.

## Competing interests

The authors declare no competing interests.
