## [Peer Review File · Nature Communications]

Cryo-EM Unveils Kinesin KIF1A's Processivity Mechanism and the Impact of its Pathogenic Variant P305LReviewers' Comments:

Reviewer #1:

Remarks to the Author:

This manuscript presents a technically outstanding, highly detailed and insightful structure/function analysis of the KIF1A kinesin motor. KIF1A is noteworthy for its exceptionally long run length ('superprocessivity') and also for its disease relevance. This manuscript makes important contributions to both aspects of this motor. While this is primarily a structure paper, the functional assays performed on the wild type and mutant motors provide exceptional, essential insights and strongly complement/support the interpretations and conclusions. It is quite striking to me that the two point mutations examined, both prolines, address long-standing questions about fundamental aspects of kinesin motility- the role of the neck linker and the K-loop in processivity. The many new details provided in the structures and other experiments will surely stimulate the field. The manuscript itself is also very comprehensive and highly polished. Well done!

Other comments:

The manuscript already seems to be very carefully edited. After some searching I finally found a grammatical correction in the supplement:

'These classes ****haven**** been designated as T1, T2 and T3' (P. 3)

Reviewer #2:

Remarks to the Author:

This paper by Benoit et al. describes high-resolution structures of various cycles of KIF1a, which is a highly processive kinesin motor protein. The most important results that would contribute to this field are the structure of the K-loop of KIF1A and its interaction with the C-terminal tails of alpha- and beta-tubulins. The authors also address another important related question regarding the motility mechanisms of KIF1A: the chemical and mechanical cycles of KIF1A in its functionally oligomerized states, including the open/closed conformations of the motor domains and neck linker. Additionally, the authors' analyses extend to the structural effects of one of the hundreds of mutations that cause KIF1A-associated neurological disorders (KANDs). The structural analyses are generally carefully done and reported to address the above questions with partial experimental support by motility assays with wild-type and mutant KIF1A proteins.

Overall, the data presented in this paper would be instrumental to the experts in this field, especially biophysicists and structural biologists who are interested in processive motor proteins and KANDs. The findings also have some potential relevance to understanding the pathophysiology of other neurodegenerative diseases. Therefore, the reviewer recommends this paper be published in Nature Communications after the following issues are resolved and re-reviewed.

Major Issues:

1. The use of ColabFold implementation of AlphaFold2 for testing the reduced binding.

The authors used ColabFold to predict the conformation of the P305L mutant. Because this is a "prediction," about the orientation of a side chain of F303, the reviewer thinks moving it from results to discussion is more appropriate. Alternatively, the authors should use molecular dynamics.

2. Figure 2d and page 7:

The plot is inappropriate because the results of the plot are used to categorize the X-axis. The reviewer suggests revising the plot to a simple 1D plot and then categorizing the dots into three categories, as shown in Supplementary Figure 5.

3. The Proposed Movement Mechanism Model (Page 19, Second Paragraph):

The authors proposed a new idea for how KIF1A moves, which is different from the recently accepted model for kinesin(-1). This is interesting, but it needs to be clarified if the 2-head bound structure in the AMPPNP state, as shown in this paper, is physiologically relevant and needs further experimental support. Recent single-molecule studies suggest that in kinesin-1, which the authors propose to have a functionally longer neck linker than KIF1A, the 2-head bound state would not be a physiological intermediate state. The authors might want to add some more cautious discussion. From this viewpoint, discussion would be deepened by citing previous reviews such as by Hirokawa's group (Nat Rev Mol Cell Biol 2009) which already discussed the interthread tension and steric restrictions based on the crystal structure of KIF1A.

Minor Issues.

1. Introduction to K-loop (Page 4, Line 3):

The Hirokawa group first discovered and named the K-loop, which is essential for the improved processivity of KIF1A through binding with the C-terminus of tubulin. This work should be referenced and discussed in this section.

2. Functional Oligomerization (Page 4, Line 11):

The Vale and Hirokawa groups' foundational work on the enhanced unidirectional processive movement of KIF1A due to functional oligomerization is crucial for understanding the current research landscape. Their key publications (Science, 2002; Nature, 2003) should be explicitly cited to acknowledge their contributions and enable readers to trace the development of this research area.

3. Functional Evidence (Page 5, Line 3):

The sentence implies that functional evidence is provided by elucidating the full structure of the K-loop and its interaction with tubulin tails. However, the evidence presented is structural rather than functional. It's necessary to accurately represent the nature of the evidence provided and make sure readers understand the insights gained from these findings.

4. Kinetics of MT-Activated ADP Release (Page 8, Line 13):

The Hirokawa group's pioneering work on the kinetics of microtubule-activated ADP release, documented in their Science 1999 publication, is critical for understanding KIF1A's mechanism. This early work should be cited to give credit to the foundational discoveries in this field and provide readers with a thorough background.

5. K-loop Swap Mutant Analysis (Page 11, Lines 2-9):

The Hirokawa group's investigation into K-loop swap mutants, particularly their PNAS 2000 study revealing the relationship between the number of positively charged residues in the K-loop, processivity, and microtubule affinity, is highly relevant. Mentioning and citing this study would acknowledge the original discovery and enrich the discussion by highlighting the intricate details of K-loop functionality.

6. Interaction of the K-loop with Tubulin Tails (Page 12):

The Hirokawa group's report on the chemical crosslinking of the K-loop to alpha- and beta-tubulins (Cell 2000) offers valuable insights into the interaction dynamics between the K-loop and tubulin tails, pertinent to this section. Discussing this study would provide a more nuanced understanding of the K-loop's role and its interaction with tubulin tails.

7. Figure 2d

It is helpful to indicate the positions of R216, A250, P14, S104, and Y105 in the inset.

8. P44: ANP two-head-bound structure

It should be ANP two-heads-bound structure.

9. Supplementary Figure 4

There are always dips around certain resolutions in the turquoise lines (kinesin). For example, the KIF1A-ADP FSC line drops around 8 Å. The reviewer suspected it was caused by the masks. Please explain.

10. Supplementary Figure 4:

The FSC curve of KIF1A P305L APO seems to be significantly different from other plots. Why?

11. Supplementary Figure 6e:

The map looks as if there is an orientation bias.

Reviewer #3:

Remarks to the Author:

Kinesins are microtubule motor proteins that are essential for many basic cell biological functions. There is a growing awareness that mutations in kinesins underly human pathologies. The neuronal kinesin KIF1A is critical for neuronal development and health and many human mutations have been identified in this motor. Here, Benoit and Rao report several high-resolution cryo-EM structures of KIF1A bound to microtubules, provide visualization of the elusive KIF1A-specific K-loop for the first time, and determine the structure of a pathogenic mutant that alters the K-loop conformation. They also provide nice evidence for a two-headed bound state in their structures and the work leads to proposed revisions in the current models for how kinesins coordinate their mechanochemical cycles during processive movement. The visualization of a two-head bound state, the binding of the tubulin tails to the motor and the K-loop are all highly novel and important results. The work is highly impactful for the kinesin and cytoskeleton field, as well as for the neuronal degeneration and structural biology fields. The work appears highly rigorous to me and would be of interest to a wide audience in the general structural biology and cell biological fields in my opinion. I support publication of the paper after the authors address the following comments.

- The authors could do a better job labeling the figures. For instance, there is no legend for the colored elements shown in Fig. 2. I think the colors correspond to the labels in Fig. 1, but it would be easier for readers to have the same legend on every figure. Conformational states termed in the text are also not labeled in the figures, such as state "T1" which is not labeled in Fig. 2C. I suggest putting clear color and state labels in each figure showing structures.
- Is it possible that the proline substitution in the neck-linker weakens dimerization, thus resulting in shorter run-lengths due to dissociation of the dimer? This possibility isn't considered, and the structure shows no major changes in neck-linker length as predicted in the hypothesis that is laid out. It seems worth considering alternative explanations for this mutant.
- In Fig. 4, the conserved 310 helix in L12 is highlighted in panels a and b, but is not mentioned in the text or legend which may lead to some confusion by readers.
- Table 1 shows the authors used 20-40uM kinesin protein to decorate microtubules. Are these numbers accurate? It is a surprisingly high concentration. At this concentration, it is somewhat surprising to me that they were able to get double-headed bound kinesin motors. It is also surprising to me that at these high concentrations, they still only see ~ 75% decoration in AMP-PNP, a high-affinity state. Can the authors comment on this in the paper? With respect to this point, I don't follow supplementary table 2. Can the authors better label the table to help the reader understand how to interpret the ratios of two versus one head bound states?
- Previously published data from this group suggests the P305L mutation dramatically impacts the landing of the ADP-KIF1A motor onto the microtubule (Lam et al. Sci. Adv. 2021). The authors also observe very low MT decoration of the mutant in ADP in this study, supporting this idea. It is surprising to me that this isn't discussed in more detail in the current manuscript.
- It is unclear to me why the F303V mutation predominantly rescues run length. Prior data mentioned above suggests the landing rate is primarily affected by the P305L mutation, which is consistent with

the tubulin clash shown in the structures presented here and mentioned here (“Consequently, this mutation likely shifts the equilibrium towards the MT-unbound (or weakly-bound) states, as we observe here.”). Therefore, I would expect removing this clash would primarily enhance the landing rate of the F303V/P305L motor, but this is not reported. Is there an enhancement of the motor landing rate as well as processivity?

- “Our structural and functional insights indicate that the K-loop’s impact on KIF1A’s superprocessivity primarily stems from dynamic electrostatic interactions between the K-loop and the MT.” As far as I know, this has been the model in the field for a long time, yet the authors seem to be suggesting it is a new idea from them.

We thank the reviewers for their careful reading of our manuscript, their positive comments, and constructive criticisms. In response to their comments, we have added new text, particularly in the introduction and discussion, to better place our results in the context of previous findings and proposals regarding the KIF1A motility mechanism. Figures and Tables have been modified according to the reviewers' suggestions, and new data and two Supplementary Figures (Supplementary Figures 10 and 11, previous Supplementary Figure 10 is now 12) were added to the manuscript .

To facilitate the review of the revised manuscript, we have highlighted the modified text with red fonts. Below, please find our responses to each of the reviewers' comments.

Reviewer 1

*'These classes ****haven**** been designated as T1, T2 and T3' (P. 3)*

Corrected.

Reviewer 2

1. The use of ColabFold implementation of AlphaFold2 for testing the reduced binding. The authors used ColabFold to predict the conformation of the P305L mutant. Because this is a "prediction," about the orientation of a side chain of F303, the reviewer thinks moving it from results to discussion is more appropriate. Alternatively, the authors should use molecular dynamic

As suggested by the reviewer, we have moved this section to the Discussion.

2. Figure 2d and page 7:

The plot is inappropriate because the results of the plot are used to categorize the X-axis. The reviewer suggests revising the plot to a simple 1D plot and then categorizing the dots into three categories, as shown in Supplementary Figure 5.

We appreciate the reviewer's comment. Accordingly, we have revised Fig. 2d as suggested.

3. The Proposed Movement Mechanism Model (Page 19, Second Paragraph):

The authors proposed a new idea for how KIF1A moves, which is different from the recently accepted model for kinesin(-1). This is interesting, but it needs to be clarified if the 2-head bound structure in the AMPPNP state, as shown in this paper, is physiologically relevant and needs further experimental support.

We appreciate the reviewer's insightful feedback and their encouragement to provide further clarification on the proposed motility model and its relationship with prior studies.

The widely accepted hand-over-hand model for the processive motion of kinesin-1 implies that a two-heads-bound state exists for at least a fraction of the mechanochemical cycle. This state is

essential for facilitating processive motion. Without it, the motor would alternate between one-head-bound and unbound states, leading to a loss of processivity. However, it's important to acknowledge that the exact duration of time spent in the two-heads-bound state relative to other states within the mechanochemical cycle remains an area of inquiry. In the discussion, we have addressed this aspect by citing relevant literature and highlighting that the relative duration of the two-heads-bound state depends on various factors, including the rates of hydrolysis, product release, trailing head detachment, and reattachment.

Regarding whether the two-heads bound structure in the AMPPNP state, as shown in this paper, is physiologically relevant, it could be argued that the use of AMPPNP rather than the physiological substrate ATP may introduce artifacts. However, AMPPNP, is a non-hydrolysable ATP analogue structurally very similar to ATP that has been used in many kinesin and other ATPase studies, and it is widely regarded as a faithful mimic of the ATP-bound state. Future structural studies using ATP, rather than AMPPNP will further clarify this issue.

Our main argument to propose a revised model is that there is no structural evidence supporting the idea that in the two-heads-bound state, ATP binding to the leading head is prevented as it is proposed in prevalent models. On the contrary, our high-resolution data of MT-KIF1A (this work) and MT-KIF14 complexes (Benoit et al., NC 2021) reveal that ATP analogues do bind to the leading head and there is nothing in the structures suggesting that ATP would be prevented from binding to the leading head of two-heads-bound state. Control of the progression of the ATPase cycle of one head by the other can be referred to as a coordinating gate because it keeps the ATPase cycle of the two motor domains out-of-phase. Our structural data indicate that there is not a coordinating gate preventing ATP binding to the leading head and instead indicate a different type of gate where the leading head is prevented from adopting the closed catalytic conformation, even when the same ATP analogue that induces the closed conformation in the trailing head is bound to the leading head.

At saturating (physiological) concentrations of ATP, we would expect that a two-heads-bound state with ATP bound to both heads, like the configuration shown in Fig. 7 (step 5), would be the dominant two-heads-bound state in kinesin's mechanochemical cycle. If ATP hydrolysis in the trailing head is fast, the prevalent two-heads-bound species would then contain ATP in the leading head and the hydrolysis products (ADP-Pi) in the trailing head. Note that the structures predict ATP hydrolysis to occur first in the trailing head as this is the one in the closed catalytic conformation while the leading head is held in the open conformation (this is the coordinating gate referred to in the previous paragraph). If the combined rates of product release and rear head detachment are fast compared to the rate of reattachment of the tethered head to the new leading position, then the two-heads-bound state would represent a smaller fraction of the mechanochemical cycle. At limiting concentrations of ATP when the rate of ATP binding to the leading head becomes slower than the combined rates of ATP hydrolysis, product release and rear head detachment, then the motor would spend a larger fraction of the cycle in a single-head-bound state with the MT-attached head with no nucleotide bound (apo) and the tethered head with ADP bound. This is the ATP waiting state (Fig. 7, step 3).

Recent single-molecule studies suggest that in kinesin-1, which the authors propose to have a functionally longer neck linker than KIF1A, the 2-head bound state would not be a physiological intermediate state. The authors might want to add some more cautious discussion.

It is difficult to address this point without knowing specifically which paper the reviewer is referencing here. However, we suspect that the reviewer may be alluding to the recent study by Wolff et al. (Science 2023). In this paper the authors apply a significant technical advance in single molecule fluorescence detection (MINFLUX) to study kinesin-1 motility. As in other studies and consistent with our model, in this paper they observe alternate two-heads and one-head-bound states during kinesin-1 motility. However, a potential point of contention with our model is that they conclude that ATP binds specifically to the one-head-bound state while in our model ATP can bind to the one-head- and the two-heads-bound states. Wolf et al. based their conclusion on the observations that the dwell time of the one-head-bound states decreases with increasing ATP concentration while the dwell time of the two-heads bound state was independent from the ATP concentration. These observations, however, are fully compatible with our model. In our model, ATP binding to the MT-bound head is essential for neck-linker docking and positioning the tethered head to the next tubulin binding site in the forward position. Without ATP, the motor remains in the one-head-bound ATP-waiting state indefinitely, and increasing [ATP] leads to shorter dwell times, as reported by Wolff and co-workers. In the two-heads-bound state, ATP can bind to the leading head (according to our model), but this is not required to exit this state and to transition to the one-head-bound state. Therefore, in our model, the dwell time of the two-heads-bound state would be predicted to be independent of the ATP concentration, as observed by Wolf and co-workers. Therefore, the observations described by Wolf et al. are compatible with our proposed model.

From this viewpoint, discussion would be deepened by citing previous reviews such as by Hirokawa's group (Nat Rev Mol Cell Biol 2009) which already discussed the interthread tension and steric restrictions based on the crystal structure of KIF1A.

We appreciate the reviewer's comment. We have now referenced and discussed in detail this review in the Discussion section. We also added a new Figure (Supplementary Fig. 11) to illustrate how our structural data differ from the hypothesis put forward in the review from the Hirokawa's group.

Minor Issues.

1. Introduction to K-loop (Page 4, Line 3):

The Hirokawa group first discovered and named the K-loop, which is essential for the improved processivity of KIF1A through binding with the C-terminus of tubulin. This work should be referenced and discussed in this section.

We have now cited Hirokawa's pioneering work and discussed it early in the Introduction.

2. Functional Oligomerization (Page 4, Line 11):

The Vale and Hirokawa groups' foundational work on the enhanced unidirectional processive movement of KIF1A due to functional oligomerization is crucial for understanding the current

research landscape. Their key publications (*Science*, 2002; *Nature*, 2003) should be explicitly cited to acknowledge their contributions and enable readers to trace the development of this research area.

We have now cited the Tomishige et al. (*Science* 2002) paper in the introduction, where KIF1A oligomerization and the 'conventional' hand-overhand type processive movement of KIF1A are first described. Additionally, we have included a citation to the Okada et al. (*Nature* 2003) paper in the Introduction.

3. Functional Evidence (Page 5, Line 3):

The sentence implies that functional evidence is provided by elucidating the full structure of the Kloop and its interaction with tubulin tails. However, the evidence presented is structural rather than functional. It's necessary to accurately represent the nature of the evidence provided and make sure readers understand the insights gained from these findings.

We appreciate the reviewer's comment. We have changed the paragraph to:

Our results highlight the full structure of the K-loop and its interaction with the C-terminal tails of both α - and β -tubulin, providing **structural** evidence for the K-loop's pivotal role in KIF1A superprocessivity.

4. Kinetics of MT-Activated ADP Release (Page 8, Line 13):

*The Hirokawa group's pioneering work on the kinetics of microtubule-activated ADP release, documented in their *Science* 1999 publication, is critical for understanding KIF1A's mechanism. This early work should be cited to give credit to the foundational discoveries in this field and provide readers with a thorough background.*

We appreciate the reviewer for bringing to our attention this omission. The pioneering work by Okada and Hirokawa (*Science* 1999) is now cited early in the introduction and also in this section.

5. K-loop Swap Mutant Analysis (Page 11, Lines 2-9):

*The Hirokawa group's investigation into K-loop swap mutants, particularly their *PNAS* 2000 study revealing the relationship between the number of positively charged residues in the K-loop, processivity, and microtubule affinity, is highly relevant. Mentioning and citing this study would acknowledge the original discovery and enrich the discussion by highlighting the intricate details of K-loop functionality*

We have added a reference to this paper in this section.

6. Interaction of the K-loop with Tubulin Tails (Page 12):

*The Hirokawa group's report on the chemical crosslinking of the K-loop to alpha- and betatubulins (*Cell* 2000) offers valuable insights into the interaction dynamics between the K-*

loop and tubulin tails, pertinent to this section. Discussing this study would provide a more nuanced understanding of the K-loop's role and its interaction with tubulin tails.

The crosslinking results in Hirokawa's group paper (Cell 2000) are now discussed and referenced in this section.

7. Figure 2d

It is helpful to indicate the positions of R216, A250, P14, S104, and Y105 in the inset.

Thank you for your suggestion. We have now included the positions of R216, A250, P14, S104, and Y105 in the inset as requested.

8. P44: ANP two-head-bound structure

It should be ANP two-heads-bound structure

Thank you for bringing this to our attention. The error has been corrected, and it now reads "ANP two-heads-bound structure" as suggested.

9. Supplementary Figure 4 There are always dips around certain resolutions in the turquoise lines (kinesin). For example, the KIF1A-ADP FSC line drops around 8 Å. The reviewer suspected it was caused by the masks. Please explain.

The small sharp dips seen in the corrected FSCs correspond to the resolution from which the phases were randomized to account for the effect that the mask would have on the FSC. To account for the effect of the mask in the estimated resolution, Relion uses a phase randomization procedure. It computes a masked FSC with the intact half maps, as well as a masked FSC using half maps with their high-resolution signal phase randomized. The corrected FSC (the ones displayed in the Supplementary Figure 4) is computed by Relion from the other two FSC. The FSC calculated directly from the half-maps (not the Relion corrected one) do not have these dips but can produce slightly overestimated $FSC_{0.143}$ values. We have chosen to show the corrected FSC curves and quote the more conservative resolution estimates derived from them.

10. Supplementary Figure 4:

The FSC curve of KIF1A P305L APO seems to be significantly different from other plots. Why?

In comparison with the other datasets the KIF1A^{P305L}-APO FSC curve shows a dip and a bump near 1/4Å. Figuring out which one(s) among the several possible factors is causing this data set to produce an FSC curve different from the others is not straightforward and it may require acquiring another dataset. We noticed that using smoother masks to calculate the FSCs reduces this effect. We have now recalculated the FSC with a smoother mask (low pass filtered to 1/20Å in the tubulin region and 1/15Å in the kinesins region). Note that the kinesin region mask is filtered at a higher frequency to minimize overestimation of the resolution in the kinesin region by including parts of the usually better resolved tubulin region. The overall $FSC_{0.143}$ resolution with the recalculated FSCs is 3.5 Å (3.4 Å before). Note that this difference in estimated resolution does not impact the final locally filtered and sharpened map.

*11. Supplementary Figure 6e:
The map looks as if there is an orientation bias.*

We appreciate the reviewer's observation, although we are unable to discern the specific indication of orientation bias in Supplementary Figure 6e. There may be some tangential elongation of the densities at higher radius from the microtubule axis caused by azimuthal mobility of the kinesin motor domain.

If the map were produced by image-projections with a limited range of angular orientations (orientation bias), we would typically expect the densities to appear elongated in the direction with less angular representations. However, we don't observe this characteristic elongation in the map. Note also that orientation bias is unlikely to occur in these reconstructions as these are helical specimens where the 3D map has been produced from a distribution of angular projections of the asymmetric unit covering the full 360 deg range (i.e. no preferential orientations or missing cone).

Reviewer #3

- The authors could do a better job labeling the figures. For instance, there is no legend for the colored elements shown in Fig. 2. I think the colors correspond to the labels in Fig. 1, but it would be easier for readers to have the same legend on every figure. Conformational states termed in the text are also not labeled in the figures, such as state "T1" which is not labeled in Fig. 2C. I suggest putting clear color and state labels in each figure showing structures.

Thank you for the suggestion. We have now added labels to Figures 1 and 2 to provide clarity for readers.

- Is it possible that the proline substitution in the neck-linker weakens dimerization, thus resulting in shorter run-lengths due to dissociation of the dimer? This possibility isn't considered, and the structure shows no major changes in neck-linker length as predicted in the hypothesis that is laid out. It seems worth considering alternative explanations for this mutant

We appreciate the reviewer's suggestion and have now mentioned the alternative explanation in the text. However, we believe this scenario to be unlikely or to have a minimal effect, considering that all the constructs include a leucine zipper to promote dimerization (as described in the Methods section), positioned further down the native neck-coil sequence and far from the mutated proline. Furthermore, the relative number of two-heads-bound cases in our cryo-EM data is similar for WT and the P364L mutant (the 2HB and 1HB values corresponding to the P364L mutant are now included in the revised Supplementary Table 2). If the P364L were to destabilize dimer formation, we would have expected either none or a significantly smaller fraction of two-heads-bound cases for the P364L mutant compared to WT.

- In Fig. 4, the conserved 310 helix in L12 is highlighted in panels a and b, but is not mentioned in the text or legend which may lead to some confusion by readers.

Thank you for bringing this to our attention. We have now added a description of the conserved 3₁₀-helix in L12 in the legend of Fig. 4. (a description of this structural element was already included in the text but later in the results section).

- Table 1 shows the authors used 20-40uM kinesin protein to decorate microtubules. Are these numbers accurate? It is a surprisingly high concentration. At this concentration, it is somewhat surprising to me that they were able to get double-headed bound kinesin motors. It is also surprising to me that at these high concentrations, they still only see ~ 75% decoration in AMPPNP, a high-affinity state.

The concentrations reported in Table 1 are indeed accurate. However, it's important to note that these concentrations refer to the concentrations of single motor chains, not dimers. Thus, the concentration of KIF1A dimers would be half of the concentrations stated. We now clarified this in the legend.

We (and others) usually use an excess of kinesin in order to improve decoration when doing cryo-EM of kinesin microtubule complexes. Even in those conditions in our experience we rarely achieve 100% decoration. Obtaining double-heads-bound kinesin motors under these conditions is not unexpected, given that once one head binds to the MT, the likelihood of the partner head binding to the nearby tubulin binding site increases due to the higher local concentration compared to the probability of another kinesin molecule binding at the same site from solution. However, as reported in Supplementary Table 2 and illustrated in Supplementary Fig. 3, we do see dimers where the partner motor domain could not bind to the MT, due to the tubulin binding site being occupied by the motor domain of another dimer. The higher the decoration, the more pronounced this 'overcrowding' effect was observed.

With respect to this point, I don't follow supplementary table 2. Can the authors better label the table to help the reader understand how to interpret the ratios of two versus one head bound states?

Thank you for the feedback. We have revised Supplementary Table 2 and its legend to provide clearer labeling and explanation on how the ratios of two versus one-head-bound states were calculated. We have also added the values corresponding to the P364L mutant dataset to this table.

- Previously published data from this group suggests the P305L mutation dramatically impacts the landing of the ADP-KIF1A motor onto the microtubule (Lam et al. Sci. Adv. 2021). The authors also observe very low MT decoration of the mutant in ADP in this study, supporting this idea. It is surprising to me that this isn't discussed in more detail in the current manuscript.

Thank you for bringing this up. The landing rates of the constructs are now reported in Supplementary Figure 10 and we have included a discussion on this topic in the manuscript.

- It is unclear to me why the F303V mutation predominantly rescues run length. Prior data

mentioned above suggests the landing rate is primarily affected by the P305L mutation, which is consistent with the tubulin clash shown in the structures presented here and mentioned here (“Consequently, this mutation likely shifts the equilibrium towards the MT-unbound (or weakly-bound) states, as we observe here.”). Therefore, I would expect removing this clash would primarily enhance the landing rate of the F303V/P305L motor, but this is not reported. Is there an enhancement of the motor landing rate as well as processivity?

Thank you for your observation. Indeed, we have found that the F303V/P305L double mutant exhibits an enhanced motor landing rate relative to the P305L mutant, which we now report in Supplementary Figure 10.

- Our structural and functional insights indicate that the K-loop’s impact on KIF1A’s superprocessivity primarily stems from dynamic electrostatic interactions between the K-loop and the MT.” As far as I know, this has been the model in the field for a long time, yet the authors seem to be suggesting it is a new idea from them.

Thank you for your feedback. We have modified the sentence as suggested to clarify that our findings align with the previous models for KIF1A motility, referencing the work of Okada and co-workers (PNAS 2000).

Reviewers' Comments:

Reviewer #2:

Remarks to the Author:

The authors have adequately addressed the reviewers' comments in their revision. They have provided thorough responses and made the necessary changes to the manuscript, significantly improving their work's quality and clarity. This reviewer commends their effort in addressing each point raised during the review process and believes that the manuscript is now ready for publication.

Reviewer #3:

Remarks to the Author:

The authors have done a nice job responding to my comments and I support publication of this revised manuscript. I believe the work will make a big impact in the kinesin field and provide some novel insight into the mechanisms of KAND.